# Spatial frequency representation in V2 and V4 of macaque monkey

**Ying Zhang[1,2], Kenneth E Schriver[1,2,3]\*, Jia Ming Hu[1,2,3]\*, Anna Wang Roe[1,2,3]\***

[1]Department of Neurosurgery of the Second Affiliated Hospital, Interdisciplinary Institute of Neuroscience and Technology, School of Medicine, Zhejiang University, Hangzhou, China; [2]Key Laboratory of Biomedical Engineering of Ministry of Education, College of Biomedical Engineering and Instrument Science, Zhejiang University, Hangzhou, China; [3]MOE Frontier Science Center for Brain Science and Brain-Machine Integration, School of Brain Science and Brain Medicine, Zhejiang University, Hangzhou, China

**Abstract** Spatial frequency (SF) is an important attribute in the visual scene and is a defining feature of visual processing channels. However, there remain many unsolved questions about how extrastriate areas in primate visual cortex code this fundamental information. Here, using intrinsic signal optical imaging in visual areas of V2 and V4 of macaque monkeys, we quantify the relationship between SF maps and (1) visual topography and (2) color and orientation maps. We find that in orientation regions, low to high SF is mapped orthogonally to orientation; in color regions, which are reported to contain orthogonal axes of color and lightness, low SFs tend to be represented more frequently than high SFs. This supports a population-based SF fluctuation related to the 'color/orientation' organizations. We propose a generalized hypercolumn model across cortical areas, comprised of two orthogonal parameters with additional parameters.

## Editor's evaluation

This article makes an important contribution to the study of early visual representation in primates by showing that intermediate cortical areas V2 and V4, as well as primary cortical area V1 (previously shown), contain orthogonal maps of orientation and spatial frequency. The authors provide convincing evidence of this fundamental principle of functional mapping across the two-dimensional cortical surface that ensures and optimizes the complete representation of these two coding dimensions.

## Introduction

Spatial frequency (SF) selectivity is a fundamental feature encoded in the visual system. Previous studies have shown that the organization of SF selectivity is related to orientation and color maps in the primary visual cortex (V1) and have a high degree of periodicity in both cats (*Hübener et al., 1997*; *Issa et al., 2000*; *Shoham et al., 1997*; *Tootell et al., 1981*) and monkeys (*Silverman et al., 1989*). Studies have consistently shown an orthogonal mapping of SF and orientation, suggesting an efficient arrangement that provides each orientation access to a wide range of SFs (*Issa et al., 2000*; *Nauhaus et al., 2012*; *Nauhaus et al., 2016*; *Xu et al., 2007*). In contrast, color representation in V1 (the color is represented in patches commonly referred to as 'blobs' in V1) is generally associated with a range of lower SFs (*Silverman et al., 1989*; *Tootell et al., 1988*). Thus, in addition to a gradient from high to low of SF representation across eccentricities (*Foster et al., 1985*; *De Valois et al., 1982*; *Lu et al., 2018*; *Yu et al., 2010*), SF organization is further specified within local distinct functional

**\*For correspondence:**
kenschriver@zju.edu.cn (KES);
hujiaming@zju.edu.cn (JMingH);
annawang@zju.edu.cn (AWangR)

**Competing interest:** The authors declare that no competing interests exist.

regions. This systematic architecture in V1 suggests that SF may be a fundamental feature of the cortical 'hypercolumn' (c.f., *Silverman et al., 1989*: organized cortical modules; *Hübener et al., 1997*: 'mosaics' of functional domains for the different properties; *Swindale et al., 2000*: uniform coverage of cortical maps).

Whether there are systematic associations between SF and other parameters in extrastriate areas, such as V2 and V4, is not known. The traditional view of the V2 hypercolumn comprises the alternating thin-pale-thick-pale stripe cycle (*Horton, 1984*; *Livingstone and Hubel, 1984*; *Roe and Ts'o, 1995*; *Tootell et al., 1983*). Within thin stripes, surface properties, typically associated with low SF preferences, such as hue maps (*Xiao et al., 2003*), 'brightness' maps (*Roe et al., 2005*), and ON/FF maps (*Wang et al., 2007*) are represented. Within the thick and pale stripes are higher-order orientation maps such as those defined by illusory contours (*Ramsden et al., 2001*), motion direction maps (*Lu et al., 2010*), and maps for motion-defined edges (*Chen et al., 2016*), as well as stereo-defined near-to-far disparity maps (*Chen et al., 2008*). Neuronal response for features such as texture have also been described (*Freeman et al., 2013*), but functional organization has not yet been investigated. There is little systematic data relating SF representation in V2 to functional stripes (cf., *Gegenfurtner et al., 1996*; *Levitt et al., 1994*; *Tootell and Hamilton, 1989*) and, despite previous attempts, few studies have demonstrated functional mapping of stripes based on SF alone (*Lu and Roe, 2007*; *Lu et al., 2018*).

In V4, surface and shape information are organized into, for lack of better terminology, 'color' and 'orientation' bands. Within the color bands, maps for hue and for luminance have been described (*Tanigawa et al., 2010*; *Liu et al., 2020*; *Li et al., 2022*). Within orientation bands, there are maps for contrast-defined contours (*Hu et al., 2020*; *Li et al., 2013*; *Lu et al., 2018*; *Tang et al., 2020*; *Tanigawa et al., 2010*), disparity-defined contours (*Fang et al., 2019*), as well as maps for curvature degree and curvature orientation (*Hu et al., 2020*; *Ponce et al., 2017*). Despite our growing understanding of functional organization in V4, how SF preference maps (first reported in *Lu et al., 2018*) relate to other feature maps in V4 remains unknown.

As part of our investigation into 'hypercolumn' organization in extrastriate cortical areas, we propose a general hypercolumn layout for V2 and V4 that includes SF (cf., *Roe et al., 2009*; *Ts'o et al., 2009*). Based on previous results reported in V1 (*Foster et al., 1985*; *Lu et al., 2018*; *Nauhaus et al., 2012*; *Nauhaus et al., 2016*; *Silverman et al., 1989*; *Tootell et al., 1988*; *Yu et al., 2010*), we predict that in each area (1) the range of SF associations shift with the topographic location of the 'hypercolumn' (*Figure 1A*), (2) orientation-selective regions (*Figure 1B*, blue) have a range of low (light gray) to high (dark gray) SF representation; iso-SF contours (purple dashed lines) map orthogonally to iso-orientation contours (green dashed lines), and (3) color-selective regions (*Figure 1B*, orange) exhibit an association with a range of low SFs (light gray). To address this proposal, we imaged V1, V2, and V4 of macaque monkey via intrinsic signal optical imaging (ISOI) with large cortical fields of view that contained sufficient territory to allow comparisons of functional organization at a range of eccentricities (*Figure 2*). Quantification of the relationship between SF maps and color and orientation maps in V2 and V4 revealed organizations that generally support our proposal for a hypercolumn architecture.

## Results

### Overall SF preference across imaged visual cortical areas

Imaging a large field of view of the cortex makes it possible to directly compare the response differences between different visual areas (*Figure 2*). *Figure 3* shows the blood vessel maps (A, E) and corresponding SF preference maps (B, F) for two separate cases wherein we imaged regions spanning V1, V2, and V4. The SF preference maps (*Figure 3B and F*) were generated by calculating the response amplitude of each pixel in these areas for stimuli with six different SFs, each at orientations of 45° and 135°. We found that in both cases, most of the imaged V1 region favored high SF stimuli, while most of the imaged V4 region favored low SF stimuli. To examine the proportion of cortical area dedicated to a single SF preference in each area (V1, V2, and V4), we calculated for each SF a coverage ratio (the area of SF-preferential response divided by the whole area, i.e., V1, V2, or V4, within the field of view). For V1, the coverage ratio peaks at high SF (light gray bars in *Figure 3C and G*), while for V4, the coverage ratio peaks at low SF (black bars in *Figure 3C and G*),

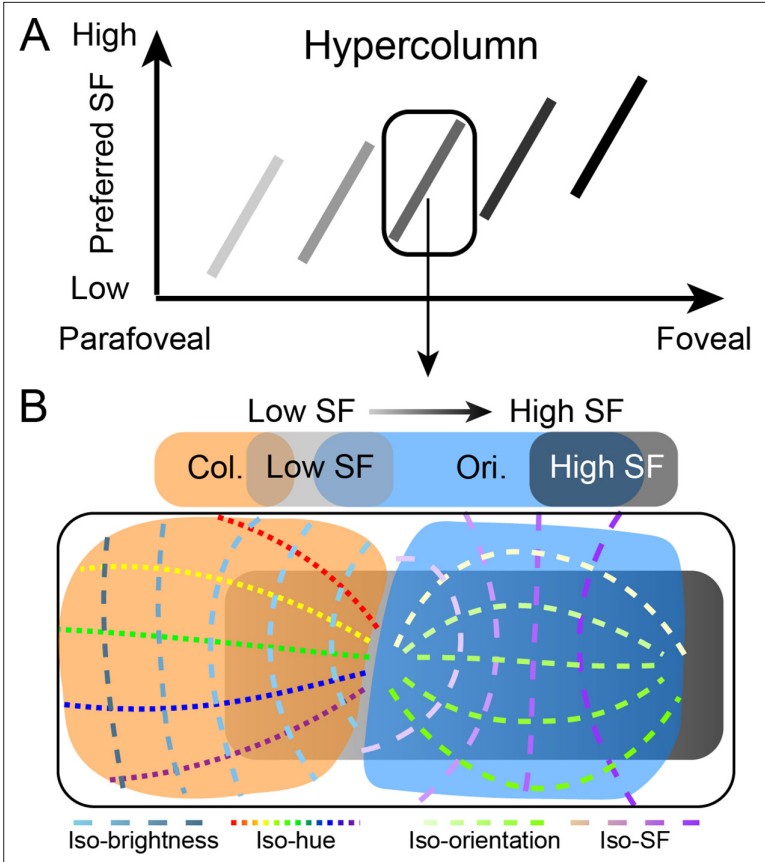

**Figure 1.** Illustration of proposed hypercolumn (including spatial frequency [SF], color, and orientation domains) in the visual cortex. (**A**) As eccentricity decreases from parafoveal to foveal region, the preferred SF gradually increases (represented as the brightness of the short bar). However, a local region (marked by a rectangle) covers a full range of SF representations in its corresponding topographic locations. This local region can be considered a 'hypercolumn.' (**B**) Details of structure in a single hypercolumn. In this local region, color domains (orange area) and orientation domains (blue area) exhibit different relationships with SF domains (light gray region: low SF preference domain; dark gray region: high SF preference domain). Orientation maps orthogonally to SF maps (green dashed lines: iso-orientation contours; purple dashed lines: iso-SF contours); an extensive range of SFs are available to each orientation. In comparison, color domains tend to have more spatial overlap with low SF preference domains and avoid overlap with high SF preference domains. In color domains, another orthogonal relationship exists between hue (dotted lines with different colors: iso-hue contours) and brightness (blue dashed lines: iso-brightness contours).

consistent with previous findings that V4 prefers lower SF than V1 (*Lu et al., 2018*). The preferred SFs (*Figure 3D and H*) significantly decrease from V1 to V4 (see *Table 1*, Kolmogorov–Smirnov test, V1 vs. V4, V2 vs. V4, V1 vs. V2, p<0.001). Our maps, in contrast to earlier studies (*Foster et al., 1985*; *Levitt et al., 1994*; *Lu and Roe, 2007*; *Lu et al., 2018*; *Silverman et al., 1989*), directly show the overall SF preference in cortical space.

To confirm that the use of just two orientations, 45° and 135°, detects a complete picture of SF preference in the imaged area, we compared the SF preference results acquired by different orientations (45° + 135°, 45°, 135°) (see *Figure 3—figure supplement 1*) and did two-way ANOVA analysis (nine values went into the two-way ANOVA, including coverage ratios of high SF regions or low SF regions in V1, V2, and V4 acquired from different orientation comparisons, 45°, 135°, 45 + 135°). We found that, for different orientations, there were no significant differences (two-way ANOVA, p>0.05) in the coverage ratios of high SF preference (SF = 4 cycles/deg) and low SF preference (SF = 0.25 cycles/deg) regions in the imaged area. In contrast, significant differences were found for different visual areas (two-way ANOVA, p<0.05).

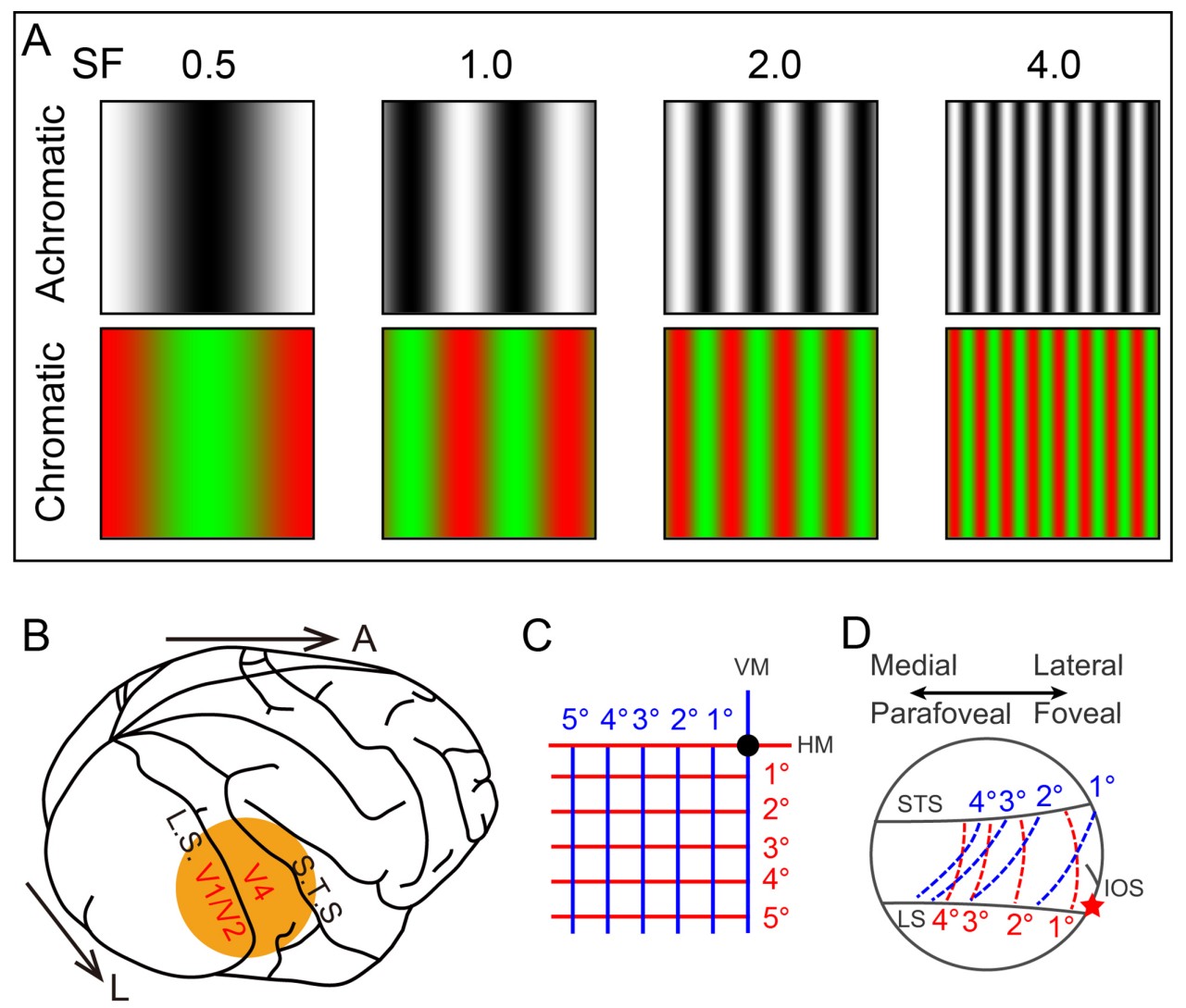

**Figure 2.** Experimental parameters. (**A**) Visual stimuli. Top and bottom rows show the black/white and green/red full screen sinusoidal gratings for four different spatial frequencies (SFs) (indicated by the numbers on top, in cycles/deg). Here for demonstration, the stimulus size is set to 2°. (**B**) Diagram of imaging site in the right hemisphere. L: lateral; A: anterior. (**C**) Lower-left visual field. Black dot: fovea. Horizontal (red) and vertical (blue) lines mapped in (**D**). (**D**) Schematic mapping of lines in (**C**) in V4 (corresponding to the orange disc in **B**). The lateral part of the imaged region corresponds to the foveal region, while the medial part corresponds to the parafoveal region. LS: lunate sulcus; STS: superior temporal sulcus; IOS: inferior occipital sulcus. Red star: estimated foveal location.

### Foveal to parafoveal shift of SF representation in V4

The large-scale imaging allowed us to capture highly structured maps of functional domains (e.g., orientation and color domains in *Figure 4A and B*, respectively) and reveal changes to those maps for different SF conditions. Below, we present our results on orientation maps and then color maps.

In the imaging results shown in *Figure 4D* (left panel), we acquired differential orientation maps using six different SFs. We found that the functional map at a specific SF is not always easily ascertained. Gratings with low SFs (e.g., 0.25 cycles/deg, 0.5 cycles/deg) evoked clear selective responses (*Figure 4D*, thresholded, right panel) in all imaged V4 regions, whereas gratings with high SFs only evoked responses in certain regions, mostly in the lateral part of the cortex (toward foveal representation, red star).

To quantify this, for each tested SF, we analyzed the spatial distribution of the orientation-selective domains (see *Figure 4D* right panel, only pixels with significantly differential response to 45° vs. 135° were included). For each orientation map obtained at a given SF, we defined

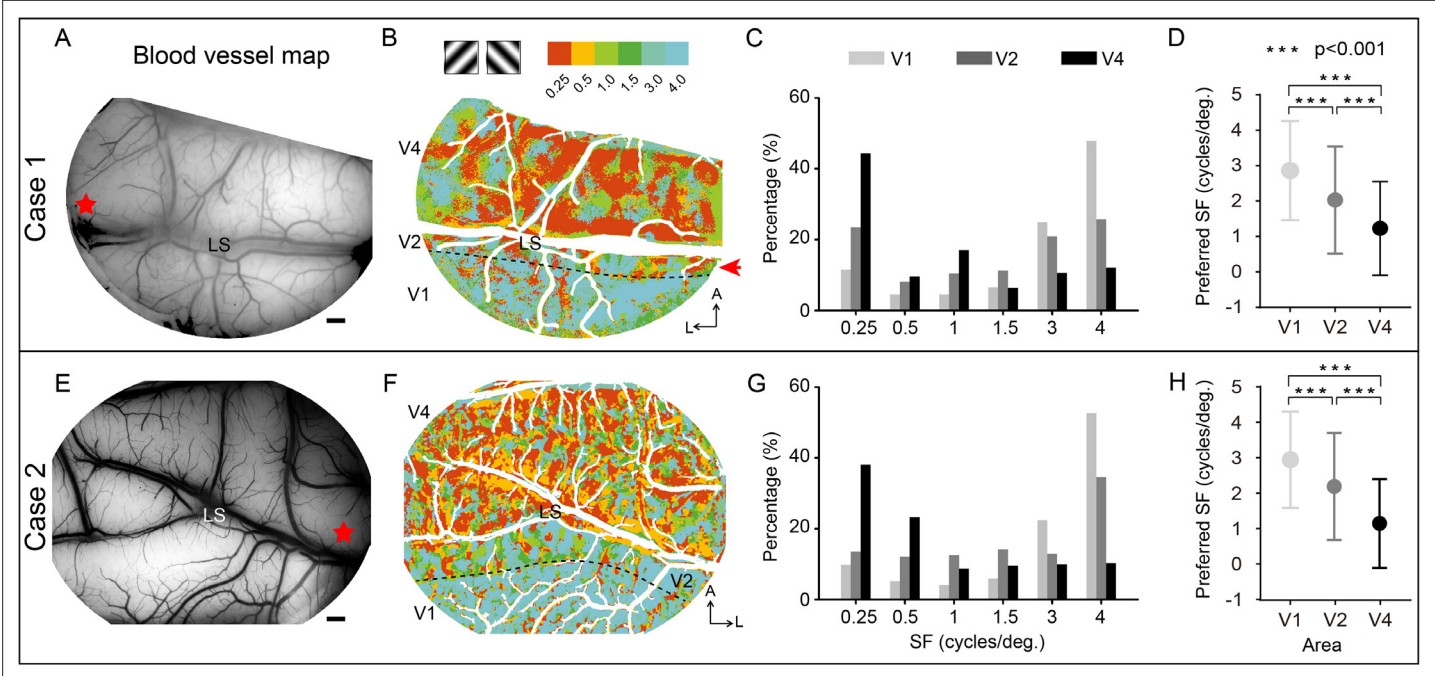

**Figure 3.** Two examples of overall spatial frequency (SF) preference in visual cortex. (**A, E**) Blood vessel map of the imaged region for cases 1 and 2, respectively. V2 and V4 are separated by the lunate sulcus (LS). Red star: estimated foveal location. Scale bar here and in all subsequent figures are 1 mm. (**B, F**) SF preference maps for cases 1 and 2, respectively. Each SF stimulus contains two orientations, 45° and 135°. For each pixel, the preferred SF is defined as the SF corresponding to its strongest response. Different colors represent different SF preferences (see color bar at top). The border between V1 and V2 (defined by ocular dominance image) is indicated by a black dashed line. A, anterior; L, lateral. (**C, G**) The coverage ratio of SF preference in each visual cortical area (light gray: V1; medium gray: V2; black: V4). (**D, H**) Mean ± SD for the preferred SF of all pixels across V1, V2, and V4. ***Kolmogorov–Smirnov test, p<0.001.

The online version of this article includes the following figure supplement(s) for figure 3:

**Figure supplement 1.** Two examples of spatial frequency (SF) preference maps obtained with different orientations (**A–F**: case 1; **G–L**: case 2).

the 'selective activation center' as the geometric centroid of all significantly activated pixels (*Figure 4C*, two-tailed *t*-test, p<0.01). As SF increases, the location of the selective activation center shifts from medial to lateral across the cortex (*Figure 4C*, blue dots). We confirmed that this remains true for each of the other cases (*Figure 4—figure supplement 1*, four cases from four hemispheres of three monkeys). Thus, the lack of response in the medial region to high SFs leads to a spatial medial-to-lateral shift of the selective activation center with increasing SF (*Figure 4—figure supplement 1B*).

We also examined color maps. Many previous V4 imaging studies employed grating stimuli to obtain color maps. However, most of these studies (*Li et al., 2014*; *Tanigawa et al., 2010*) have not addressed how SF affects color selective response. To test this, we recorded cortical responses to red/green isoluminance sinusoidal gratings with six different SFs and compared these responses with those to achromatic stimuli of the same SF (*Figure 4E*). Color domains that showed significantly stronger responses (two-tailed *t*-test, p<0.01, N = 30) to red/green stimuli were marked in black (*Figure 4E*, right panel). Similar to orientation domains, under low SF conditions

**Table 1.** Summary of the preferred spatial frequencies (SFs) in different visual areas.

| Preferred SF (mean ± SD cycles/deg.) | V1 | V2 | V4 |
|---|---|---|---|
| Case 1 | 2.86 ± 1.40 N = 147,207 | 2.03 ± 1.51 N = 103,377 | 1.23 ± 1.32 N = 490,921 |
| Case 2 | 2.96 ± 1.37 N = 194,847 | 2.20 ± 1.52 N = 186,288 | 1.15 ± 1.26 N = 635,679 |

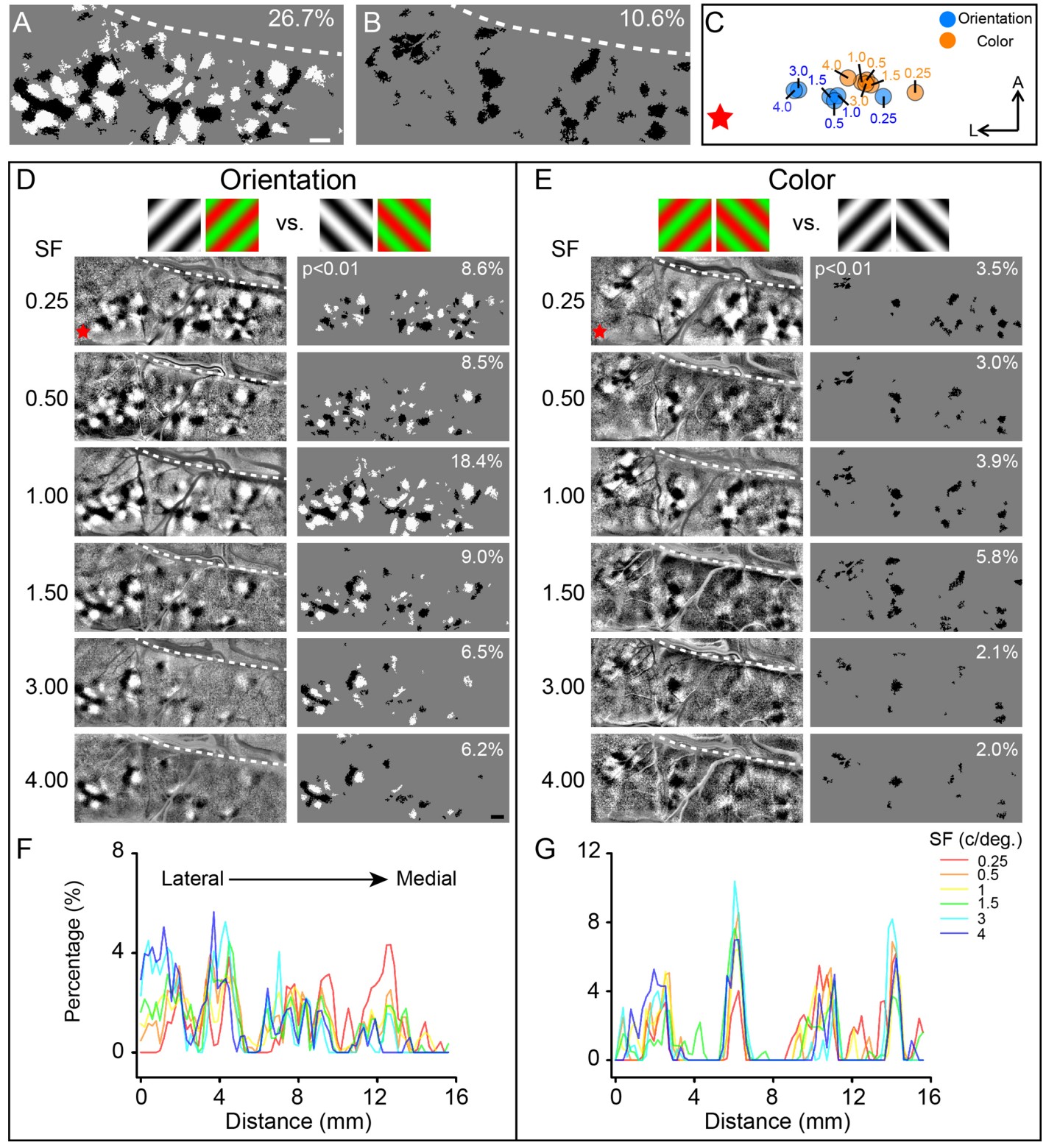

**Figure 4.** Comparison of the functional maps obtained at different spatial frequencies (SFs). (**A, B**) Combined results generated by superimposing pixels in (**D**) and (**E**), respectively. Scale bar, 1 mm. (**C**) Selective activation centers of the activated orientation domains (blue dots) and color domains (orange dots) under different SF conditions. The dots are the geometric centroid of all corresponding activated regions in V4 (two-tailed *t*-test, p<0.01). The values indicate the SF used. (**D**) V4 orientation maps and corresponding activated regions for different SFs. The gratings above the maps indicate the subtraction pair for the maps. Left panel: differential maps in response to 45° (black patches) versus 135° (white patches); right panel: stimulus-

*Figure 4 continued on next page*

*Figure 4 continued*

activated orientation-selective regions, only pixels that can distinguish 45° (black pixels) from 135° (white pixels) are included (two-tailed *t*-test, p<0.01), numbers in the top-right corner indicate the coverage ratio of activated regions in the imaged V4. Red star: estimated foveal location. (**E**) Color maps and corresponding activated regions for different SFs, acquired from the same case in panel (**A**). Gratings above the maps indicate the subtraction pair for the maps. Left panel: differential maps in response to R/G gratings (corresponding to the black patches) versus W/B gratings (corresponding to the white patches). Right panel: activated color preference regions for the stimuli, only pixels showing significantly stronger responses to R/G gratings (black pixels) are included (two-tailed *t*-test, p<0.01). (**F, G**) Activated area histograms along the M-L axis generated from (**D**) and (**E**), respectively.

The online version of this article includes the following figure supplement(s) for figure 4:

**Figure supplement 1.** Orientation maps obtained by using drifting gratings with different spatial frequencies (SFs).

**Figure supplement 2.** Another example of V4 color map acquired with different spatial frequencies (SFs).

**Figure supplement 3.** The matrices show the correlation values for pairs of maps acquired with different spatial frequencies (SFs).

(<1 cycle/deg), color domains were detected in the medial region of the cortex. At high SF, the color-selective response is no longer easily discernible in the medial region (SF = 3, 4 cycles/deg, *Figure 4E*, right panel). But in the lateral cortical region, color-selective response was detected regardless of SF. Thus, the activation center of color-selective response also shifted from medial to lateral with increasing SF (*Figure 4C*, orange dots). In a second case, we obtained similar results (*Figure 4—figure supplement 2A*) and found the activation center shifted laterally as SF increased (*Figure 4—figure supplement 2B*). We found that, when only a few SFs are tested, the coverage ratio of the orientation and color domains is underestimated (see white numbers in upper corners in *Figure 4A, B, D and E*), underscoring the importance of testing a wide range of SFs.

In addition to extracting the activation centers for different SFs in V4, we also examined the spatial distributions for each SF separately. We plotted the proportion of the activated pixels at different distances from lateral (distance = 0 mm) to medial under different SF conditions (see *Figure 4F and G*). As reported in previous V4 studies (*Tanigawa et al., 2010*; *Li et al., 2013*), orientation domains and color domains tend to separate in space, forming different functional bands. As shown in *Figure 4F and G*, at distances of 2, 6, 10, and 14 mm, the percentages of activated orientation regions decrease while the percentage of activated color regions increases. We found that for the lateral orientation and color bands (see the two bands < 6 mm), the percentage values were higher for high SFs (blue/cyan lines vs. red/orange lines). In comparison, for the medial orientation and color bands (see the two bands >6 mm), the percentage values were higher for low SFs (red/orange lines vs. blue/cyan lines).

We found surprisingly that the relative proportions of high to low SF preferences differ for color and orientation. As shown in *Figure 5*, these proportions differ markedly in foveal vs. parafoveal locations. For each SF, we divided the number of activated pixels in the lateral and the medial parts (see *Figure 5—figure supplement 1*) by the total number of the activated pixels preferring this SF (pixels selectively activated with a single SF). We found that the proportions of activated pixels in color and orientation domains in the lateral and medial parts of V4 change with SFs in distinct ways. In parafoveal cortex (medial), for both orientation and color domains, the proportion of pixels tends to decrease with increasing SF; in foveal cortex (lateral), for both orientation and color domains, the proportion of pixels increase as SF increases.

The exposed areas of V1 are in the lateral region of the hemisphere, corresponding to the eccentricity of 0–2°; here low SFs barely evoke measurable selective orientation responses (*Figure 4—figure supplement 1A*, left column, cases 1–3). The orientation map in the lateral region is only apparent at high SFs (*Figure 4—figure supplement 1A*, >1 cycle/deg). Although the optimal SF differs between V1 and V4, for both areas, the selective activation center moves from medial to lateral as SF increases (*Figure 4—figure supplement 1*, cases 1–4, dots of different shading correspond to the activation centers at different SFs). In monkey visual cortex, the foveal regions of V1, V2, and V4 are located in the lateral part of the cortex (*Gattass et al., 1981*; *Gattass et al., 1988* see *Figure 2D*), in agreement with the findings from electrophysiology studies that foveal regions favor high SF (*Perry and Cowey, 1985*; *Schein, 1988*; *Wässle et al., 1989*; *Wilder et al., 1996*).

To further confirm the spatial relationship of functional maps acquired at different SFs, we calculated the cross-correlation values of these functional maps acquired at different SF conditions. Each response map was divided into two halves: foveal region, the left half close to V4 foveal region; and parafoveal region, the right half away from foveal region. The correlation values of the

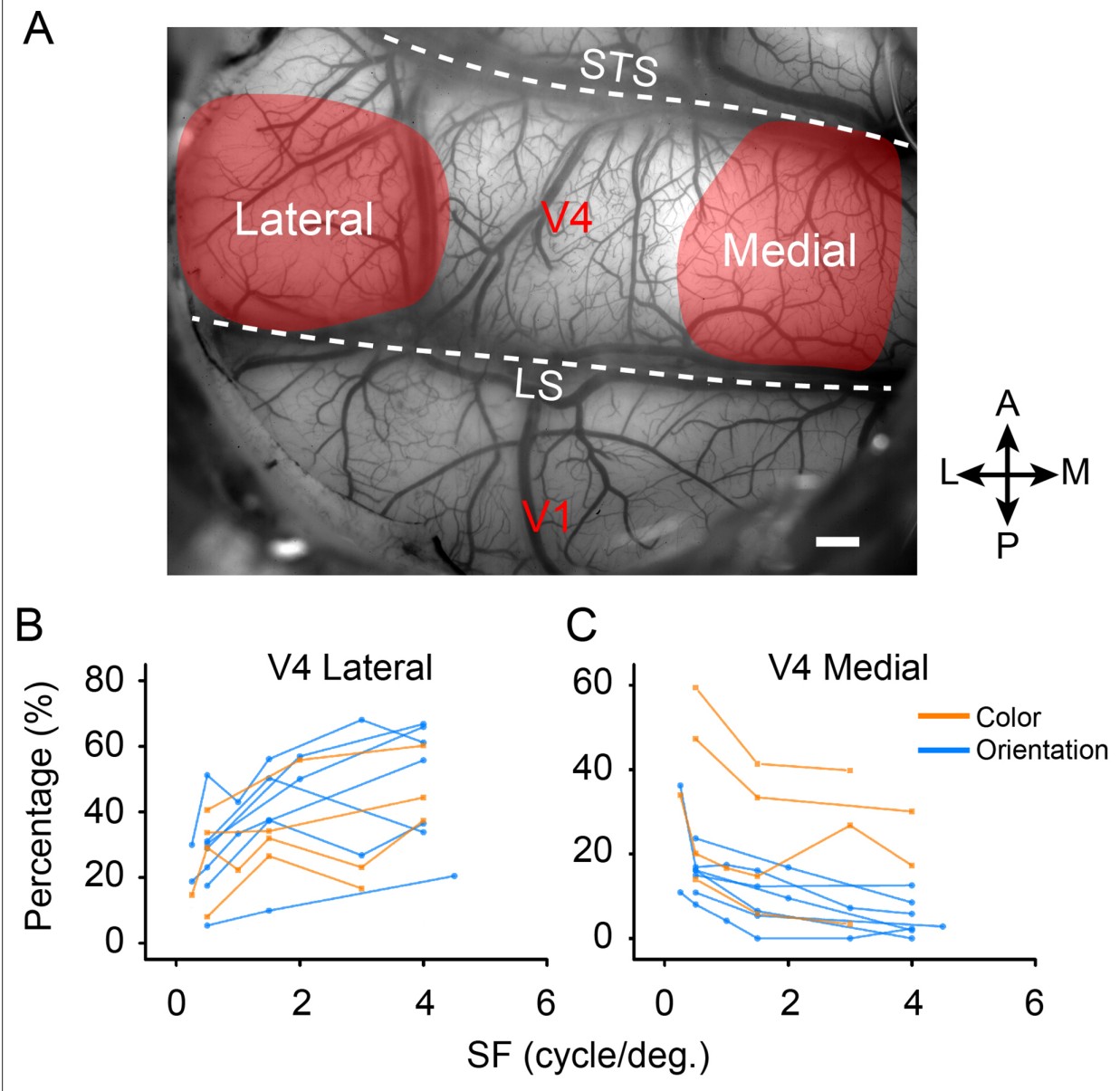

**Figure 5.** The percentage of selectively activated domains change according to spatial frequency (SF). (**A**) Demonstration of cortical blood vessel map and the visual areas chosen for analysis. See *Figure 5—figure supplement 1* for details. Scale bar, 1 mm. (**B, C**) The proportions of activated functional domains in the lateral and medial parts of V4 (color: orange, from four experiments in three hemispheres; orientation: blue, from seven experiments in five hemispheres) change according to SFs. Points connected by a line represent results from the same experiment.

The online version of this article includes the following figure supplement(s) for figure 5:

**Figure supplement 1.** Cortical blood vessel maps of all cases and the corresponding chosen areas for activated percentage analysis in V4.

two halves were calculated separately (*Figure 4—figure supplement 3A and B*, left matrix: foveal region; right matrix: parafoveal region). For orientation maps (*Figure 4—figure supplement 3A*), high correlation values (>0.5) appeared in the comparisons among high SF conditions in foveal region. In contrast, high correlation values appeared in the comparisons among low SF conditions in parafoveal region. For color functional maps (*Figure 4—figure supplement 3B*), higher correlation values were measured under low SF conditions in both foveal and parafoveal regions, while high correlation values appeared only in the foveal region under high SF conditions.

These differences reflect distinct capabilities of different cortical regions (fovea vs. parafovea). Independent of the type of visual information presented (orientation or color), parafoveal regions

tend to process the visual input containing relatively low SF components, while visual input containing high SF components is better processed in foveal regions. In V4, the foveal region is capable of processing visual information with a broad range of SF sensitivity.

## Relationship between SF and orientation maps in V4 and V2

Having obtained orientation and SF preference maps from the same cortical region, it becomes possible to analyze the spatial relationships between these maps. To identify the regions which are highly selective for orientation, for each pixel, we calculated normalized orientation values (from 0 untuned to 1 highly tuned) and set a threshold of 0.5 or larger (see orientation selectivity maps in *Figure 6B* and *Figure 6—figure supplement 2B*; and the selectivity thresholded orientation preference map in *Figure 6C*). We determined the iso-orientation and iso-SF contours based on the smoothed orientation preference map (see *Figure 6A*) and SF preference map (see example in *Figure 6—figure supplement 1*, 18 iso-orientation gradient contours and 5 iso-SF gradient contours). In V2 and V4, the iso-orientation and iso-SF contours that intersected at large angles were found most frequently (see *Figure 6G, K and O*, *Figure 6—figure supplement 2G and K*). To demonstrate that the intersection angles are more frequently detected at a large angle, we divided the detected intersection angles into three groups (small: 0–30°; medium: 30–60°; large: 60–90°) and compared the percentage difference among these groups. The results indicate that there are more (percentage value) 60–90° intersection angles than other kinds of intersection angles in the orientation-selective regions (see white patches in *Figure 6F, J and N*). The percentage of the large angle group is significantly higher than the small (Wilcoxon rank-sum test, p=1.60 × $10^{-5}$, n = 15 from five regions, two V2 regions, and three V4 regions) and medium groups (Wilcoxon rank-sum test, p=1.73 × $10^{-4}$). In addition, we compare the distribution between groups with strong orientation selectivity (e.g., *Figure 6D–O*) and weak orientation selectivity (see *Figure 6P–U*). The percentage difference in large intersection angle (60–90°) is also significant (strong orientation selectivity group, n = 15 from five regions, two V2 regions, and three V4 regions; weak orientation selectivity group, labeled as '60–90° Ref.' in *Figure 6V*, n = 6 from two V4 regions; Wilcoxon rank-sum test, p=0.0057, see *Figure 6V*).

## SF bias in V4 color domains

Since the first report on V4 SF preference domains (*Lu et al., 2018*), the relationship between V4 SF domains and V4 color domains remains unclear. It should be noted that SF preference *domains* (i.e., preference for low vs. high, *Figure 7B and C*) are distinct from SF preference *maps* (i.e., six-value SF maps, *Figure 3B and F*), as the domains are determined by statistical analysis (two-tailed *t*-test, p<0.01) and effectively distinguish high SF from low SF. Here, by comparing cortical responses recorded using high SF stimuli at two orientations: 45°, 135°, and low SF stimuli (at the same two orientations), we generated differential SF maps (see *Figure 7A*). A distinct segregation of light and dark regions is visible in this functional map. The black patches are the regions that prefer high SF to low SF, while the white patches have the opposite preference.

We compared the spatial relationship of color domains with SF domains (*Figure 7A–D*). *Figure 7B* shows the low SF preference domains color coded in gray relative to the color domains (orange patches). Similarly, high SF preference domains are color coded in gray relative to the same domains in *Figure 7C*. In this case, low SF preference domains tend to co-localize with V4 color domains to a greater extent than high SF preference domains (21.3% vs 8.3% of all pixels in color domains).

Additional examples are shown in *Figure 7—figure supplement 1* (three cases from three different hemispheres). In these four cases, we found that the total amount of cortex able to discriminate high SF vs. low SF (high SF plus low SF pixels) is only a small portion of the total V4 area (coverage ratio of high SF preference domains in V4: 3.9% ± 1.5%, mean ± SD; coverage ratio of low SF preference domains in V4: 6.0% ± 3.7%, mean ± SD.; coverage ratio: the area of SF domains divided by the imaged area of V4). Although the difference is not statistically significant (four cases, Wilcoxon test, p>0.05), in all cases, we found a great tendency for low SF domains to overlap with color domains (the percentage of high SF preference regions in color domains: 3.9% ± 3.3%; the percentage of low SF preference regions in color domains: 13.3% ± 8.4%, *Figure 7D*).

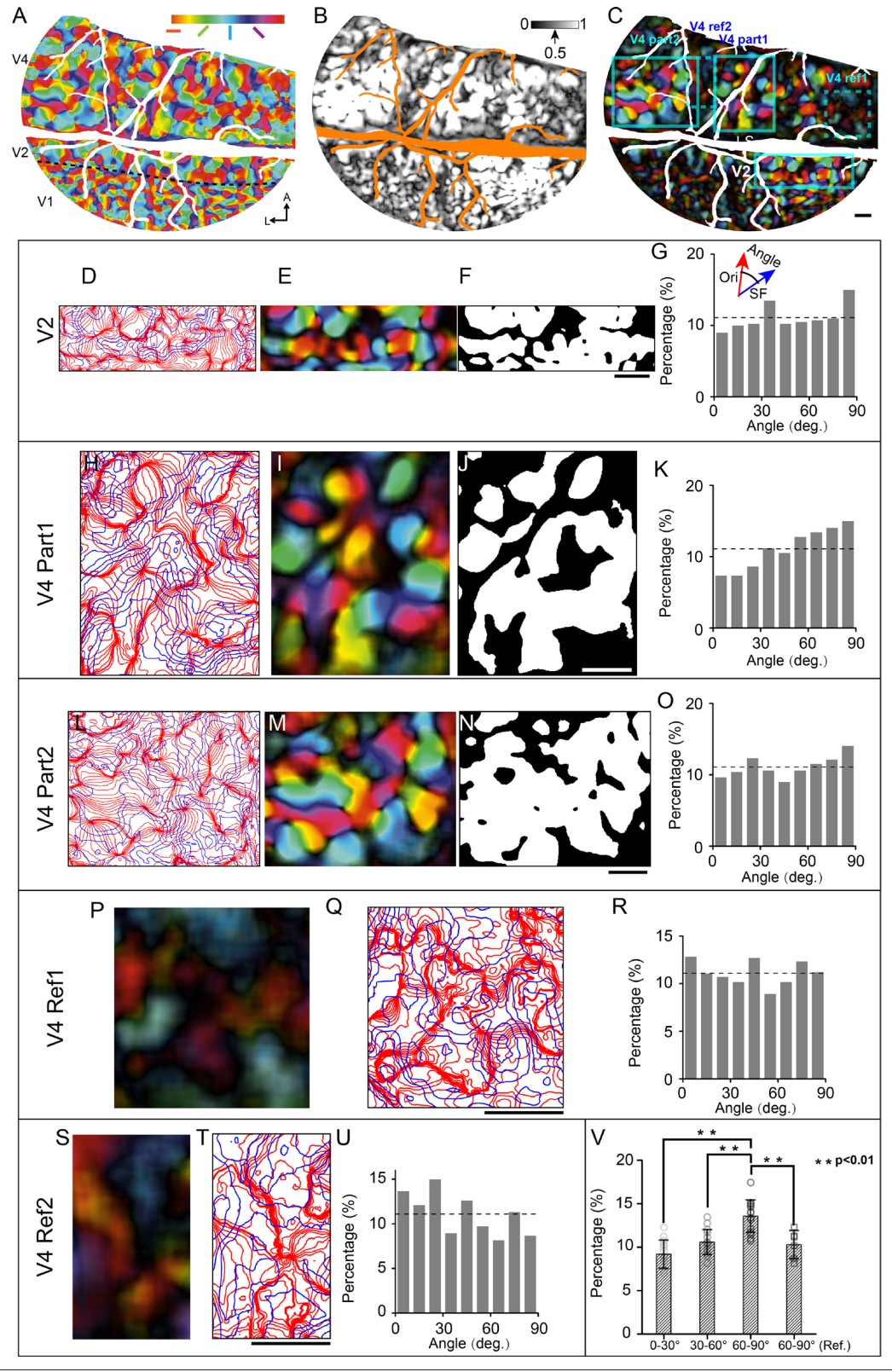

**Figure 6.** Relationship between spatial frequency (SF) and orientation maps in V4 and V2. (**A**) Orientation preference map. Different colors represent different orientation preferences. (**B**) Orientation selectivity map. The gray scale represents the normalized orientation selectivity (0: no orientation selectivity; 1: strong specific selectivity to one single orientation). (**C**) Selectivity thresholded orientation preference map (combined result from

*Figure 6 continued on next page*

*Figure 6 continued*

**A** and **B**). Cyan boxes indicate the chosen regions for intersection angle distribution analysis: one V2 region (**D–G**), two V4 regions (region 1: **H–K**; region 2: **L–O**), and two V4 reference regions with weak orientation selectivity (dotted box, V4 Ref1: **P–R**; V4 Ref2: **S–U**). (**D, H, L, Q, T**) Iso-orientation (red lines) and Iso-SF gradient contours (blue lines). (**E, I, M, P, S**) Selectivity thresholded orientation preference maps corresponding to (**D, H, L, Q, T**). (**F, J, N**) Regions (white parts) with high orientation selectivity (normalized orientation selectivity > 0.5) selected for calculating the intersection angle. (**G, K, O, R, U**) Distributions of intersection angles of the selected regions. The dashed lines indicate the expected value (11.1%) if angles are distributed randomly. (**V**) Percentage comparison (Wilcoxon rank-sum test) among different angle groups (0–30°, 30–60°, and 60–90° of strong orientation-selective regions, n=15, and 60–90° of weak orientation-selective regions: 60–90° ["Ref."], n=6). Error bar: SD. Scale bars: 1 mm.

The online version of this article includes the following figure supplement(s) for figure 6:

**Figure supplement 1.** Demonstration of intersections of iso-contour lines for orientation and spatial frequency (SF) maps.

**Figure supplement 2.** The second case of the relationship between spatial frequency (SF) and orientation maps in V4 and V2.

## Change of SF preference in V2

Previous studies in V2 have shown that relying on SF to distinguish different types of stripes (thin, pale, thick) is difficult (*Levitt et al., 1994*; *Lu et al., 2018*; *Tootell and Hamilton, 1989*). However, one study did report a stripe specific SF selectivity (*Lu and Roe, 2007*). We hypothesize that this controversy is due to the inability of cytochrome oxidase staining to reliably identify stripe type, which functional imaging can securely address. To further explore this, we first examined whether there were measurable SF preference differences in V2. As indicated in the map, the SF preference changed across V2 (same as V2 in *Figure 3B*, the region between lunate sulcus and V1/V2 border indicated by red arrow). We adopted the same method as Lu (*Lu et al., 2018*) to obtain a differential SF map (*Figure 7F*). The low SF preference regions obtained using these two methods (white patches, in the differential SF map, see *Figure 7F*; red/orange regions, in the colored SF preference map, see *Figure 7I and J*) are well correlated with color preference domains (red dashed circles in *Figure 7F* and black patches in *Figure 7G and H*). In addition, we acquired differential SF maps by subtractions between different SF pairs (see *Figure 7—figure supplement 2*). For subtraction between high SF (>1 cycle/deg) and low SF (0.5 cycles/deg) conditions, high SF preference domains can be detected in the foveal region (red star, see black patches indicated by red arrows in *Figure 7—figure supplement 2A and B*). However, for the subtraction between medium SF (1 cycle/deg) and low SF (0.25 cycles/deg) conditions, the relatively high SF preference domains can only be detected in the parafoveal region (see black patches indicated by blue arrows in *Figure 7—figure supplement 2C*). These results indicate from parafoveal to foveal region, the preferred SF tends to increase gradually.

To provide an additional illustration of how SF preference varies along V2, we selected V2 regions that exhibited two separated color domains (*Figure 7E*, regions 1 and 2) and compared the changes of SF preference against color selectivity along the V2 long axis (parallel to the black arrows in *Figure 7G and H*). Both SF preference (red lines) and color selectivity (black lines) vary along the V1/V2 border (*Figure 7G–L*). Color selectivity was found in or near low SF preference regions (see also *Figure 7F*, red dashed circles and white patches in V2); however, not all low SF preference regions exhibited strong color selectivity. Thus, SF preference differences vary uniquely within V2 (see *Figure 7M*). The differences in SF preference between the regions on the two sides of a thin stripe (see *Figure 7K and L*, thin stripe location: around the peaks of the black lines) may correlate to the reported two types (medial vs. lateral) of pale stripes in V2 (*Felleman et al., 2015*).

## Discussion

Using optical imaging of intrinsic signals in a large area of visual cortex, we were able to simultaneously characterize the functional architecture underlying color, orientation, and SF domains in areas V2 and V4. We found (1) *Topography*: With respect to topography, the SF population response of V4 orientation and color domains shift systematically with topographic location. To be specific, the geometric centroids of the selective response shift toward foveolar (lateral) parts of the cortex as SF

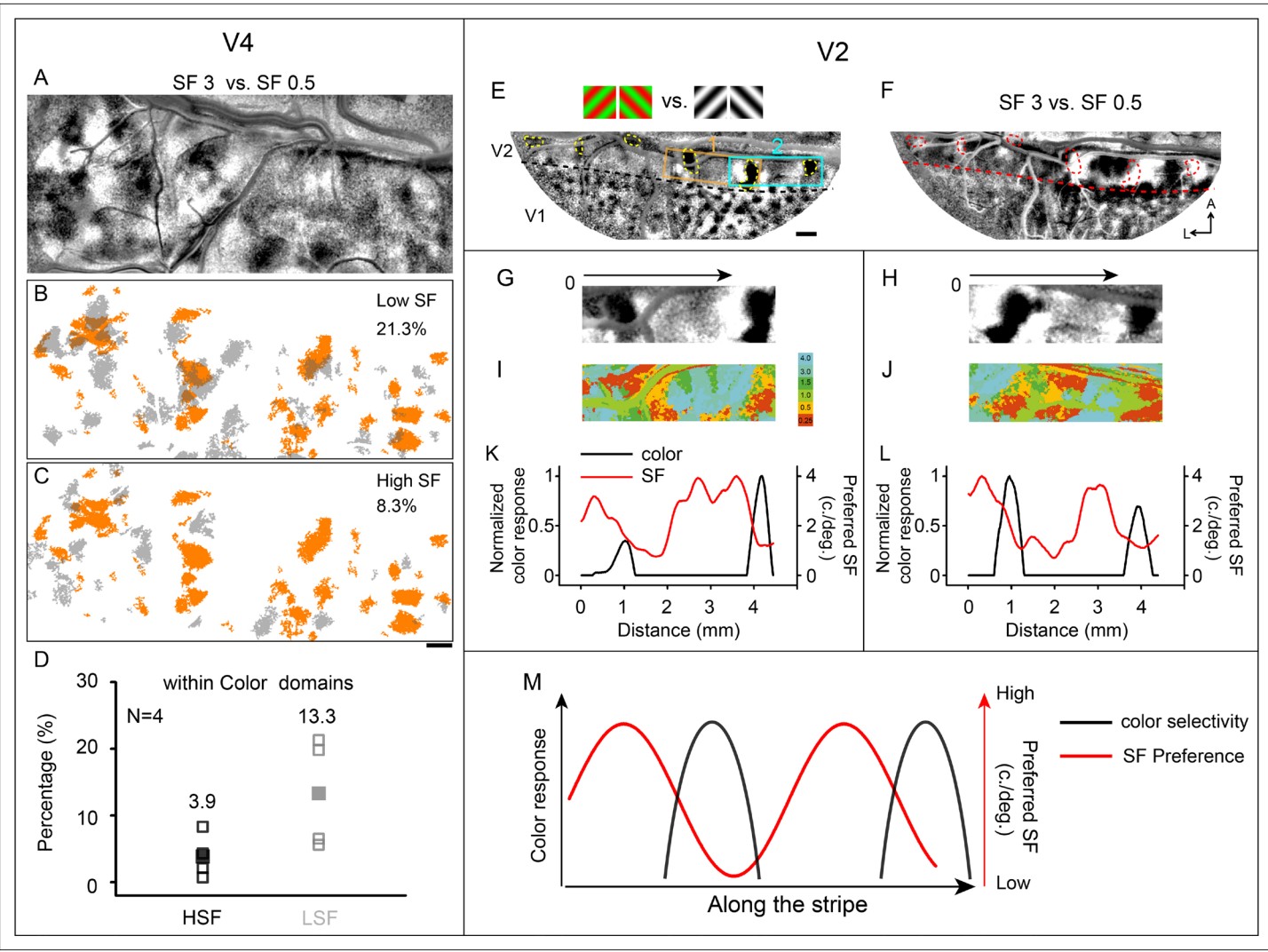

**Figure 7.** Relationship between spatial frequency (SF) and color-selective domains. (**A–D**) Relationship in V4 that high SF domains tend to avoid color domains. (**A**) Same case as in *Figure 4A*. Differential SF map in V4 is produced by subtracting the average image of two oriented grating stimuli at a low SF (0.5 cycles/deg) from the corresponding average image at a high SF (3 cycles/deg). The dark patches correspond to regions that prefer higher SF, while the white patches prefer lower SF. (**B, C**) Overlay of color domains (orange) and SF domains (gray). (**B**) Low SF domains; (**C**) high SF domains. Scale bar, 1 mm. (**D**) The percentage of HSF/LSF (high spatial frequency/low spatial frequency) selectivity regions within color domains was calculated. Unfilled squares represent the results from each case (four cases). Filled squares are the averaged outcomes from the four cases. The mean value is shown on top of the corresponding group. For the other three cases, see *Figure 7—figure supplement 1*. (**E–M**) Relationship in V2 that stripe-like distribution of SF preference changes periodically. Data from the same case shown in *Figure 3*, case 1. (**E**) Color map. Regions 1 (orange rectangle) and 2 (cyan rectangle) were selected for further analysis in (**G–L**). In V2, the yellow dashed outlines highlight the color domains. The border between V1 and V2 is indicated by a black dashed line. Scale bar, 1 mm. (**F**) Differential SF maps produced by subtracting the average image of two oriented grating stimuli at a low SF (0.5 cycles/deg) from the corresponding average image at a high SF (3 cycles/deg). Red dashed outlines: color domains same with those in (**E**). (**G, H**) Enlarged color maps from regions 1 and 2. (**I, J**) Enlarged SF maps from regions 1 and 2. (**K, L**) Changes of color-selective response (black lines) and SF preference (red lines) along the path parallel to V1/V2 border in V2. (**M**) Similar to color selective responses in V2, SF preference changes along V1/V2 border.

The online version of this article includes the following figure supplement(s) for figure 7:

**Figure supplement 1.** Relationships between V4 spatial frequency (SF) and color-selective domains in three cases.

**Figure supplement 2.** Differential spatial frequency (SF) maps acquired by subtractions between different SF pairs (same case as *Figure 7*).

increases. This finding quantifies, at a population level, the gradation of SF preference across the visual field. Interestingly, distinct from V1 fovea, which was selectively responsive to orientations of higher SFs (see *Figure 4—figure supplement 1A*), the foveal region of V4 was responsive to a broad range of SF from high to low, suggesting greater SF integration at higher cortical levels, even in foveal

regions. (2) *Orthogonal primary axes*: Within V4 and V2, similar to what was previously shown in V1, the gradients of orientation and SF maps tend to be orthogonal to one another. (3) Low *SF bias in V4 and V2 color domains*: We find there is a tendency for low SF preference domains to overlap with color domains in V4 and V2.

## General organization rules of cortical space

As mentioned in many previous studies, orthogonal crossings between different cortical maps facilitate the maximum combination of response properties in a local area with columnar organization. This kind of spatial relationship has now been reported in different animals (cat, monkey) and between different functional maps including orientation vs. ocular dominance, orientation vs. SF in cat area 17 (*Hübener et al., 1997*); orientation vs. ocular dominance, orientation vs. SF in monkey V1 (*Bartfeld and Grinvald, 1992*; *Nauhaus et al., 2012*; *Obermayer and Blasdel, 1993*); hue vs. lightness in macaque V1 (*Li et al., 2022*); orientation vs. disparity in macaque V2 (*Chen et al., 2008*; *Ts'o et al., 2009*) and, in this study, orientation vs. SF in macaque V2 and V4. These examples suggest that to effectively use cortical space, this orthogonality is established by key visual attributes.

We suggest that this orthogonality may be a common principle and reveals key parameters specific to each cortical area. Thus, for object structure, orientation and SF are two key parameters; for color, the key parameters are hue and luminance (*Li et al., 2022*). These combinations ensure a complete representation of basic shape and surface information at each cortical level. Thus, cortical mosaics contain distinct regions of orthogonal feature parameters, as observed in the color and orientation stripes of V2 and the color and orientation bands of V4, and may present for other parameters (e.g., face space, object space) in higher cortical areas.

## Population-selective responses across the cortex

The ISOI method enables us to study the population-selective responses across a large area of the visual cortex. Consistent with previous findings (*Desimone and Schein, 1987*; *Lu et al., 2018*), our results indicate that the majority of V4 regions favor low SF relative to V1 (*Figure 3*). For both V4 orientation- and color-selective response, there is a tendency for representation to shift to higher SFs from parafovea to fovea (*Figure 4* with three supplements, and *Figure 5* with one supplement), supporting the inverse relationship between SF preference and retinal eccentricity (*Desimone and Schein, 1987*; *Lu et al., 2018*). However, it should be noted that in foveal V4 region, even at low SF, robust orientation- and color-selective responses are detected. This points to an important difference between V1 and V4. That is, foveal representation in V4 may be better organized for processing complex images (e.g., natural scenes) with multiple SF components.

## SF preference organization in V2

We explored whether SF is spatially organized relative to the stripes in V2. We found that similar to color-selective response, SF preference changes within the exposed V2 area, forming different SF preference patches, which supports a general functional layout for SF coding in the visual system (preference for low SF in V1 blob, V2/V4 low SF domains; preference for high SF in V1 interblob, V2/V4 high SF domains). Based on the resolution of ISOI, we cannot achieve cellular level resolution. We cannot do further analysis in regions containing neurons with complex selectivity (e.g., orientation pinwheel). These tiny structures are best studied with other methods (e.g., two-photon imaging). More evidence and new techniques (e.g., ultra-high-field 7T fMRI) could also be introduced to test whether the SF preference in the entire V2 changes periodically as other features (e.g., color, orientation, direction, disparity).

Why does SF preference in V2 change in this way? As suggested by modeling V2 retinotopic maps of tree shrews (*Sedigh-Sarvestani et al., 2021*), in elongated cortical areas such as V2 there tend to be periodic changes in response features across the cortical surface. Another putative functional implication of this periodic distribution is to aid in the integration across features spaces (wiring minimization) via horizontal connections in V2 (*Chklovskii et al., 2002*; *Cowey, 1979*; *Hubel and Wiesel, 1977*; *Mitchison, 1991*; *Koulakov and Chklovskii, 2001*).

## Thoughts about the hypercolumn

Based on the above findings, we suggest a nested hierarchy of organizations (see *Figure 1*). At the scale of visual field representation, there is a broad and downward shifting range of SFs from center to periphery. Within this global SF map lie hypercolumns of repeated orientation and color representation, each of which contains two orthogonally arranged primary parameters. In the 'orientation' regions, SF is systematically and orthogonally mapped in relation to the orientation map (this study); in the 'color' regions, hue and lightness are orthogonally mapped (*Li et al., 2022*). Analogous to how the primary parameter spaces mapped in each cortical area change from one area to another (e.g., V1: ocular dominance, orientation, color; V2: higher-order orientation, hue, disparity; V4: curvature, 3D shape from shading, hue and lightness, *Hu et al., 2020*; *Srinath et al., 2021*; face areas: face maps, *Kanwisher et al., 1997*, *Chang and Tsao, 2017*; object areas: object maps, *Bao et al., 2020*), we hypothesize that subregions of a cortical hypercolumn also organize for different parameters. Thus, while SF is an important primary axis in orientation regions, in color regions which by nature are associated with low SFs, the rationale for a systematic SF map is weakened. One could view the color regions as evolutionary 'add-ons,' which became tacked on to the low end of the SF continuum. It should be noted that the functional maps or regions (e.g., orientation, color) in V2 and V4 are not simple repeats of the functional representation in V1, although here they have the same name (orientation, color). These maps or regions are dedicated to processing different visual information in different visual areas. We suggest the architecture of SF representation, which describes distinct SF representations within orientation and color regions, further extends and supports the view of continued parallel streaming of feature-specific pathways.

# Materials and methods

Data was acquired from five hemispheres of three adult macaque monkeys (one male and two female, *Macaca mulatta*). All procedures were performed in accordance with the National Institutes of Health Guidelines and were approved by the Zhejiang University Institutional Animal Care and Use Committee (approval no. ZJU20200022 and ZJU20200023).

## Animal preparation

Chronic optical windows were implanted in contact with the cortex above areas V1, V2, and V4, containing lunate sulcus and superior temporal sulcus as described previously (*Hu et al., 2020*, also see *Figure 2B*). The eccentricity of the visual field corresponding to exposed V4 was 0–5° and for V1/V2 was 0–2°. Following the craniotomy surgery, optical images were acquired in order to generate basic functional maps. Monkeys were artificially ventilated and anesthetized with propofol (induction 5–10 mg/kg, maintenance 5–10 mg/kg/hr, i.v.) and isoflurane (0.2–1.5%). Anesthetic depth was assessed continuously by monitoring heart rate, end-tidal $CO_2$, blood oximetry, and EEG. Rectal temperature was maintained at 38°C. Animals were paralyzed (vecuronium bromide, induction 0.25 mg/kg, maintenance 0.05–0.1 mg/kg/hr, i.v.) and respirated. Pupils were dilated (atropine sulfate 1%) and eyes fitted with contact lenses of appropriate curvature to focus on a stimulus screen 57 cm from the eyes.

## Visual stimuli for optical imaging

Visual stimuli were created using ViSaGe (Cambridge Research Systems Ltd.) and displayed on a calibrated 27-inch monitor (Philips 272G5D) operating at 60 Hz refresh rate. The luminance for white stimuli was 206.52 cd/m² and black was 0.50 cd/m². Full-screen visual grating stimuli were used to locate different functional domains. To acquire color maps (*Figure 4E*, *Figure 4—figure supplement 2*, *Figure 7B, C and E*, *Figure 7—figure supplement 1*), red/green isoluminance (red: CIExyY, 0.662, 0.328, 40; green: CIExyY, 0.320, 0.613, 40) and black-white sine-wave drifting grating stimuli, as shown in *Figure 2A*, were presented at two different orientations (e.g., 45° and 135°) with various SFs. To acquire orientation maps (differential orientation maps: *Figure 4D* and *Figure 4—figure supplement 1*; orientation preference maps: *Figure 6A*, *Figure 6—figure supplement 1A*, *Figure 6—figure supplement 2A*; orientation selectivity map: *Figure 6B*, *Figure 6—figure supplement 2B*; selectivity thresholded orientation preference map: *Figure 6C*, *Figure 6—figure supplement 2C*) and SF maps (differential SF maps: *Figure 7A and F*, *Figure 7—figure supplement 2*; SF preference maps:

*Figure 3B and F*, *Figure 3—figure supplement 1A–C, G–I*, *Figure 6—figure supplement 1B*, *Figure 7I and J*), gratings with different orientations (0°, 45°, 90°, 135°) and different SFs (0.25, 0.5, 1, 1.5, 3, 4 cycles/deg) were presented. The temporal frequency of the gratings was fixed to 4 Hz, and the corresponding drifting speeds of these SF conditions (0.25, 0.5, 1, 1.5, 3, 4 cycle/deg) are 16, 8, 4, 2.7, 1.3, 1 deg/s. The different directions of motion were randomly interleaved.

**Table 2.** Comparisons used to generate different functional domains.

| Comparison | Domain type | ΔdR/R criteria |
|---|---|---|
| RG vs. WB | Color | <0 |
| | Luminance | >0 |
| 0° vs. 90° | 0° | <0 |
| | 90° | >0 |
| 45° vs. 135° | 45° | <0 |
| | 135° | >0 |
| High SF (≥2 cycles/deg) vs. low SF (<1.5 cycles/deg) | High SF | <0 |
| | Low SF | >0 |

SF: spatial frequency.

## Optical imaging

The brain was imaged through a glass window mounted in contact with cortex. Images of cortical reflectance changes (intrinsic hemodynamic signals) corresponding to local cortical activity were acquired (Imager 3001, Optical Imaging Inc, German Town, NY) with 630 nm illumination. Image size was 1080 × 1308 pixels representing 14.4 × 17.4 or 8.7 × 10.5 mm field of view. Visual stimuli were presented in a random order. Each stimulus was presented for 3.5–4.5 s. Frames were acquired at 4 Hz for 4–5 s synchronized to respiration. Visual stimuli were presented 0.5 s after beginning image acquisition. The imaging data were stored in a block fashion. Each block contained the imaging data recorded from the stimulus conditions (presented one time). Each stimulus was presented at least 25 times.

## Data analysis

### Generation of functional maps

With the following formula, $\Delta R_i = (\bar{R}_{i1} - \bar{R}_{i2}) \times \sqrt{N}/S_i$, we assessed the response differences between two comparison groups. $\bar{R}_{i1}$ and $\bar{R}_{i2}$ are the mean dR/R values ($dR/R = \frac{R_{9-end} - R_{1-3}}{R_{1-3}}$, $R_{9-end}$ is the averaged response from frames 9 to the last frame, $R_{1-3}$ is the averaged response from frames 1–3) in the two compared conditions for pixel i, N is the number of trials, and Si is the standard deviation of $(R_{i1} - R_{i2})$. Single condition maps were obtained by comparing the images acquired during stimulus and during a blank.

Color maps were obtained by comparing red/green and white/black grating images, differential SF maps were obtained by comparing high (2–6 cycles/deg) and low (0.25–0.5 cycles/deg) SF images, and differential orientation maps were obtained by comparing two orthogonal orientation images (45° vs. 135°). Maps were low-pass filtered (Gaussian filter, ~30–80 μm diameter) and low-frequency noise was reduced by convolving a given map with a~1–2 mm diameter Gaussian filter and subtracting from the original map. Within a single experimental session, the same filtering parameters were always used to ensure that this filtering procedure did not influence the observed differences. The border between V1 and V2 was determined based on color map: in V1 color response has a blob-like distribution, whereas in V2 color response has a stripe-like distribution.

To generate SF preference maps (e.g., in *Figure 3B and F*), for each pixel we compared its activation under different single SF conditions. The preferred SF of each pixel was defined as the SF at which the strongest activation signal (amplitude averaged from frame 10 to frame 20 in each condition) for that pixel was observed. The comparison includes two orientations (45° and 135°); for each orientation, six different SFs, 0.25, 0.5, 1, 1.5, 3, and 4 cycles/deg, were presented. Each pixel in a given SF preference map was assigned a unique color to represent the preferred SF. Orientation preference maps (*Figure 6A*, *Figure 6—figure supplement 2A*) were calculated based on single orientation condition maps (four orientations, 0°, 45°, 90°, 135°), and each pixel was assigned a unique color to represent the preferred orientation (*Bosking et al., 1997*).

## Locating the positions of selective activation and determining the activation center

Functional domains were identified by selecting the pixels with a significant difference in dR/R (two-tailed *t*-test, p<0.01) under two comparison conditions (see *Table 2*).

For a given activated region, the activation center was defined as the geometric centroid of all significantly activated pixels within the region; for example, the orientation-selective activation center in *Figure 4D* is the centroid of all the pixels of 45° and 135° orientation domains under one SF condition and the color activation center in *Figure 4E* is the centroid of all the pixels of color domains under one SF condition. The overlap between different functional domains was also calculated based on these thus-defined functional domains (see *Figure 7B–D*, *Figure 7—figure supplement 1*).

## Calculating the correlation of pairs of maps

To quantify the correlation between two functional maps, we isolated the significant responses in the imaged area of V4 (regions that were significantly activated by the visual stimuli, two-tailed *t*-test, p<0.01) and calculated the cross-correlation values between the maps (*Figure 4D and E*, right panels) acquired under different SF conditions. To compare the difference between foveal and parafoveal regions, we divided the imaged V4 regions into two halves: the left half of the region was designated as foveal, the right half was designated as parafoveal, and the correlation value for each half was calculated separately.

## Comparing the spatial relationship between SF and orientation maps

Iso-orientation contours (18 contours, 5°, 15°, 25°, 35°, 45°, 55°, 65°, 75°, 85°, 95°, 105°, 115°, 125°, 135°, 145°, 155°, 165°, and 175°) and iso-SF contours (five contours, 0.25, 0.5, 1, 2, and 3 cycles/deg) were drawn based on these smoothed maps using the MATLAB 'contour' function. We quantified the orientation selectivity of each pixel by calculating the vector sum of the responses to the four tested orientations (0°, 45°, 90°, and 135°). The length of each vector was normalized to a range of 0–1 by dividing the largest vector length. We calculated the difference between the two gradients at each intersection within the strong orientation-selective regions (normalized orientation selectivity > 0.5, e.g., *Figure 6*, V2 and V4 parts) or weak orientation-selective regions (normalized orientation selectivity < 0.5, e.g., *Figure 6*, V4 refs) to determine the spatial relationship between SF and orientation maps.

## Characterizing the layout of SF preference in V2

As reported in previous studies, V2 color-selective response changes periodically along the long axis of V2 (*Levitt et al., 1994*; *Roe and Ts'o, 1995*). To characterize the periodic change of SF preference in V2, we chose a region of V2 with clearly identifiable periodic changes in color response (at least two well-separated color domains) for further analysis. We slightly rotated the selected V2 region to align the V1/V2 border horizontally in the cropped small map (see *Figure 7G and H*). For each of these small maps, we quantified and normalized color selectivity and SF preference for all pixels in the map. The average value for the pixels along each vertical line at different distances from left (distance = 0 mm) to right (distance = 4.5 mm) was then computed and plotted.

## Calculating the coverage ratio of selective activation

We first determined the activated orientation and color domains to calculate the proportion of activated regions in a given area. The coverage ratio (e.g., the values at the corners in *Figure 4D and E*) was calculated by dividing the number of the selectively activated pixels recorded at each SF by the total number of pixels in the given area. To evaluate the weights of the activated pixels at different distances from lateral (distance = 0 mm) to medial (*Figure 4F and G*), we calculated by dividing the number of the selectively activated pixels along each vertical line at different distances by the total number of activated pixels with the given SF. To evaluate the response weights of V4 lateral and medial parts in different SFs (*Figure 5B and C*) at each single SF condition, we

calculated the ratio of the activated pixels in lateral and medial parts to all the activated pixels for a given SF.

After merging all the orientation- and color-selective pixels at different SF conditions (at least three SFs, low: 0.25–0.5 cycles/deg; medium: 1–2 cycles/deg; and high: 3–6 cycles/deg), we obtained the regions representing nearly the entirety of orientation/color domains (see example in *Figure 4A and B*) and calculated the coverage ratio of the functional domains within the area (values in *Figure 4A and B*).

## Acknowledgements

We thank Yin Liu and Meizhen Qian for help with the animal experiments. This research was conducted at Zhejiang University and was supported by the China Brain Initiative (grant no. 2021ZD0200401 to AWR), National Key R&D Program of China (grant no. 2018YFA0701400 to AWR), the National Natural Science Foundation of China (grant nos. 31627802, U20A20221, and 81961128029 to AWR; grant no. 32100802 to JMH), China Postdoctoral Science Foundation (grant no. 2020M681829 to JMH), and the MOE Frontier Science Center for Brain Science & Brain-Machine Integration, Zhejiang University.

## Additional information

### Funding

| Funder | Grant reference number | Author |
|---|---|---|
| China Brain Initiative | 2021ZD0200401 | Anna Wang Roe |
| National Key Research and Development Program of China | 2018YFA0701400 | Anna Wang Roe |
| National Natural Science Foundation of China | 31627802 | Anna Wang Roe |
| National Natural Science Foundation of China | U20A20221 | Anna Wang Roe |
| National Natural Science Foundation of China | 81961128029 | Anna Wang Roe |
| The MOE Frontier Science Center for Brain Science & Brain-Machine Integration | | Anna Wang Roe |
| National Natural Science Foundation of China | 32100802 | Jia Ming Hu |
| China Postdoctoral Science Foundation | 2020M681829 | Jia Ming Hu |

The funders had no role in study design, data collection and interpretation, or the decision to submit the work for publication.

### Author contributions

Ying Zhang, Data curation, Formal analysis, Investigation, Writing - original draft, Writing – review and editing; Kenneth E Schriver, Formal analysis, Supervision, Writing – review and editing; Jia Ming Hu, Conceptualization, Data curation, Formal analysis, Supervision, Funding acquisition, Validation, Investigation, Visualization, Methodology, Writing - original draft, Project administration, Writing – review and editing; Anna Wang Roe, Conceptualization, Resources, Supervision, Funding acquisition, Writing - original draft, Project administration, Writing – review and editing

### Author ORCIDs

Ying Zhang ⓘ http://orcid.org/0000-0001-9631-8280
Jia Ming Hu ⓘ http://orcid.org/0000-0002-5306-445X
Anna Wang Roe ⓘ http://orcid.org/0000-0003-4146-9705

## Ethics

All procedures were performed in accordance with the National Institutes of Health Guidelines and were approved by the Zhejiang University Institutional Animal Care and Use Committee (Permit Number: ZJU20200022 and ZJU20200023).

## Decision letter and Author response

Decision letter https://doi.org/10.7554/eLife.81794.sa1
Author response https://doi.org/10.7554/eLife.81794.sa2

# Additional files

## Supplementary files

• MDAR checklist

## Data availability

All data generated or analysed during this study are included in the manuscript and supporting file. All data used in the figures have been deposited at Open Science Framework (https://osf.io/agkr7/).

The following dataset was generated:

| Author(s) | Year | Dataset title | Dataset URL | Database and Identifier |
|---|---|---|---|---|
| Hu J | 2022 | Spatial frequency representation in V2 and V4 of macaque monkey | https://osf.io/agkr7/ | Open Science Framework, agkr7 |

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
