## [Editor Report]

This article makes an important contribution to the study of early visual representation in primates by showing that intermediate cortical areas V2 and V4, as well as primary cortical area V1 (previously shown), contain orthogonal maps of orientation and spatial frequency. The authors provide convincing evidence of this fundamental principle of functional mapping across the two-dimensional cortical surface that ensures and optimizes the complete representation of these two coding dimensions.

---

## [Decision Letter]

**Decision letter after peer review:**

Thank you for submitting your article "Spatial frequency representation in V2 and V4 of macaque monkey" for consideration by *eLife*. Your article has been reviewed by 3 peer reviewers, one of whom is a member of our Board of Reviewing Editors, and the evaluation has been overseen by Tirin Moore as the Senior Editor. The following individual involved in the review of your submission has agreed to reveal their identity: Charles E Connor (Reviewer #1).

Essential revisions:

1. There are five major claims. First is a replication of lower spatial frequency representation in V4. This is based on two examples shown in Figure 3. The differences look clear but should be analyzed statistically.

2. The second claim is that, on the large scale of the visual field representation in V2 and V4, spatial frequency is mapped from high to low going from fovea to periphery, here estimated as lateral to medial, as in V1. The analysis for this is to plot the geometric centroids of 6 different spatial frequency band responses, for orientation contrasts (Figure 4A) and color contrasts (Figure 4B), and show that they progress in position from lateral to medial for high to low frequencies. This seems like an unusual analysis that obscures most of the original data concerning the relationship of spatial frequency response profiles to the two-dimensional imaging area. And, the centroids do not show a continuous map, but rather a bunching of points except for the extremes. The additional data are 4 supplemental examples with three, different frequency ranges. These data are not analyzed statistically.

3. The third and most important claim is that spatial frequency and orientation are mapped orthogonally (and recursively) in V2 and V4, as seen in Figure 5 and the Figure 5 supplement. Together these figures present two regions in V4 and two regions in V2. If these are the only analyzed regions, the authors need to specify more clearly how they were selected. Presumably, though, other regions were analyzed, and the authors should present results from all analyzable regions, and use statistical analyses to establish significance.

4. The fourth claim is that color-sensitive regions in V4 are more associated with low spatial frequencies. The one significant example (the analysis and statistical tests need to be explained), shown in Figure 6, shows a weak relationship to color for both spatial frequency bands, and the other examples presented in the supplementary are not significant and have even lower absolute relationships. These results, if presented, should be considered inconclusive.

5. The fifth claim is for stripe-like periodicity of spatial frequency representation in V2, related to color tuning. This is supported by ostention to binary maps of spatial frequency tuning in Figure 7 and supplement. Establishing this periodicity would require statistical analysis, and in any case, seems impossible since only a sliver of V2 is visible in these brain surface images, so stripes orthogonal to the V1/V2 boundary (i.e. CO stripes) cannot be distinguished from other patterns of spatial frequency tuning. In fact, Figure 5E and S5I do not appear to have iso-frequency contours biased toward that orientation.

6. I recommend emphasizing the conclusion that spatial frequency and orientation are mapped orthogonally in V2 and V4, and presenting analyses of all analyzable imaged regions, with statistical tests.

7. The analysis of spatial frequency selectivity in V1 vs. V2 vs. V4 should include a statistical test. The authors might consider a newly available explanation for lower frequency tuning in V4, that the predominant fraction of V4 neurons is tuned for 3D shape from shading, which by its nature is low in spatial frequency (Srinath et al., Current Biology, 2021). This tuning is mapped into patches that are intercalated with patches tuned for 2D shapes, and the authors could consider how this 2D/3D mapping relates to the mapping they observe for spatial frequency and orientation.

8. If the authors want to present an analysis of spatial frequency as a function of laterality, a more continuous analysis like marginal response histograms for spatial frequency along the ml axis would be more informative and amenable to statistical analysis.

9. I recommend eliminating the analyses of and claims about color vs. spatial frequency and stripe-like periodicity because the data do not seem adequate for testing the hypotheses.

10. The first concert has to do with the use of just two orientation values, 45 and 135 degrees to form a neutral orientation control condition (a cocktail blank), where (I assume because the details were sketchy) responses are added, and then subtracted from the comparison condition with a different color or spatial frequency. Surely, horizontal and vertical orientation regions of the cortex will not be stimulated, or weakly at best, and it will be harder or impossible to determine the preferences of these regions for other parameters, such as SF or color. If the authors are sure this is not a problem I think it needs to be addressed directly, early on in the paper.

11. The second concern has to do with the distribution of iso-parameter contour intersection angles (e.g. Figures5d and h). The results seem, literally, too good to be true. Previous studies (e.g. Obermeyer and Blasdel, 1993; Hubener et al., 1997) have all shown very much broader distributions. It makes me wonder if the cocktail blank problem above has resulted in some kind of systematic distortion of the SF map, such that the intersection angles are biased (I do not have a more developed perspective on it than that). The second possibility is some kind of numerical error and I would urge the authors to check their code – e.g. try the same calculation with an OR map from one animal superimposed on an SF map from another and see if the expected random distribution is found.

12. I will add that the finding of decreasing SF preference with increasing distance from the fovea seems unremarkable and I would suggest de-emphasising it as a main result. It is certainly relevant to interpreting the observations of course because SF can only be interpreted with respect to eccentricity. But it is hardly a new or unexpected finding. Also, the fact that color processing is restricted to low SF values is well established in the color literature – this is probably an unavoidable consequence of the distribution of red and green cones across the retina. Some reference to these long-established results might be made.

13. I found the proposed hypercolumn architecture in Figure 1B very difficult to understand. SFs vary in a continuum, so why are only two levels (low SF and high SF) shown in two different colors? Iso-orientation and Iso-SF lines could have been shown in different colors also (say HSV colors for orientation to show the circular mapping and gray colormap for SF going from low to high). Similar to what has been done for iso-hue and iso-brightness lines in the color region. Perhaps it may be worthwhile to show the same proposed architecture in V1 as well, in which orientation maps form pinwheels and colors are in separate blobs. It was unclear to me how this architecture could have pinwheels/blobs as well.

14. It is not clear to me how the details of the functional maps depend on the choice of stimuli. In single unit studies, typically a large number of orientations and SFs are used to independently map the SF and orientation tuning preferences. In contrast, here only 2 orientations are used in one case to map the color space. Even for mapping the orientation space, only 4 orientations are used. For mapping the color space also, only the hues along the red-green axis are varied (L-M pathway). I understand that some of these choices could be due to the recording modality (imaging), but it would be very useful if the authors could discuss how/if these stimulus choices can affect their results. More details of the stimuli, such as the drift rate of the gratings, and the cie (x,y, Y) coordinates of red and green hues would be useful.

15. Can you show the iso-contour lines for orientation on the orientation maps also as a supplementary figure to see how well the algorithm works? Figure 5A shows iso-orientation lines on the SF map. The iso-SF contours shown in Figure 5B easily correspond to the colors in the SF map shown in 5A, but I had difficulty mapping the orientation. Also, I was wondering whether the way the comparisons are done to get the maps (for example, in Figure 4, the same 4 stimuli are compared in two different ways to get orientation and color maps) can potentially impose some constraints on those maps. I say this because it is striking to me that almost every red and blue line shown in 5C and 5G appears to intersect orthogonally (as also shown in 5D and 5H).

16. To me the orthogonality of SF and Orientation contours in Figure 5 was the most striking result. Can you show how this analysis looks for V1? The supplementary figure also shows only V2 and V4.

17. The claim about periodicity is not well quantified. If the authors wish to make this claim, they need to show the Fourier transform of the activation pattern as a function of space and show clear peaks in the spectrum. Also, the authors can perhaps clarify what is the spatial resolution of the imaging technique itself.

---

## [Author Response]

Essential revisions:1. There are five major claims. First is a replication of lower spatial frequency representation in V4. This is based on two examples shown in Figure 3. The differences look clear but should be analyzed statistically.

We thank the reviewers for this suggestion to include statistical analysis. The mean and standard deviation of preferred spatial frequencies in V1, V2, and V4 have now been calculated for both cases. In addition, we have tested whether the differences in distributions among these areas are significant. These statistics have been added to Figure 3 and the Results section.

Lines 106-115:

“The preferred SFs (Figure3D, H) significantly decrease from V1, to V4 (Case1: V1 mean preferred SF = 2.86 cycles/deg., SD=1.40 cycles/deg., N=147207; V2 mean preferred SF = 2.03 cycles/deg., SD=1.51 cycles/deg., N=103377; V4 mean preferred SF = 1.23 cycles/deg., SD=1.32 cycles/deg., N=490921; Case2: V1 mean preferred SF = 2.96 cycles/deg., SD=1.37 cycles/deg., N=194847; V2 mean preferred SF = 2.20 cycles/deg., SD=1.52 cycles/deg., N=186288; V4 mean preferred SF = 1.15 cycles/deg., SD=1.26 cycles/deg., N=635679; Kolmogorov-Smirnov test, V1 vs. V4, V2 vs. V4, V1 vs. V2, p<0.001).”

2. The second claim is that, on the large scale of the visual field representation in V2 and V4, spatial frequency is mapped from high to low going from fovea to periphery, here estimated as lateral to medial, as in V1. The analysis for this is to plot the geometric centroids of 6 different spatial frequency band responses, for orientation contrasts (Figure 4A) and color contrasts (Figure 4B), and show that they progress in position from lateral to medial for high to low frequencies. This seems like an unusual analysis that obscures most of the original data concerning the relationship of spatial frequency response profiles to the two-dimensional imaging area. And, the centroids do not show a continuous map, but rather a bunching of points except for the extremes. The additional data are 4 supplemental examples with three, different frequency ranges. These data are not analyzed statistically.

We thank the reviewer for raising this question. As we claimed in the previous manuscript, we found that the activated V4 orientation domains and V4 color domains progress in position from fovea to periphery when SF is mapped from high to low, as in V1. Due to the small exposure of the V2 area, we did not show how V2 orientation/color maps change according to different SFs.

Figure 4A (orientation) and 4B (color) depict how cortical activations gradually shift with different SFs in the large imaged field. Many details are in the image results. The centroid calculations depict the general tendency of the changes. In addition to extracting the geometric centroid, we now have provided one example of how the activated region percentage (See New Figure 4C and 4D and the related description in the Results and Methods sections) changes along the M-L axis for evaluating these shifts. It details how activated domains change with topological eccentricity under each SF condition. Furthermore, we have summarized our data and provided the population tendency of how activated functional domains (orientation and color domains) gradually change in fovea vs. parafovea (see New Figure 5).

Lines 155-167:

“In addition to extracting geometric centroid with different SFs, we also draw the weights of the activated pixels at different distances from lateral (distance=0 mm) to medial under different SF conditions (see Figure 4C, D). As reported in previous studies (Tanigawa et al., 2010; Li et al., 2013), orientation domains and color domains tend to separate in space, forming different functional bands (see Figure 4C, D, at distances of 2, 6, 10, and 14 mm, the percentages of activated orientation regions decrease while the percentage of activated color regions increase). We found that for the lateral orientation and color bands (see the two bands with a distance smaller than 6 mm), the percentage values are higher in high SF conditions (blue/cyan lines vs. red/orange lines). In comparison, for the medial orientation and color bands (see the two bands with distances larger than 6 mm), the percentage values are higher in low SF conditions (red/orange lines vs. blue/cyan lines).

Lines 575-593:

“Calculating the coverage ratio of selective activation.

We first determined the activated orientation and color domains to calculate the proportion of activated regions in a given area. The coverage ratio (e.g., the values at the corners in Figure 4 A, B) was calculated by dividing the number of the selectively activated pixels recorded at each SF by the total number of pixels in the given area. To evaluate the weights of the activated pixels at different distances from lateral (distance=0 mm) to medial (Figure 4C, D), we calculated by dividing the number of the selectively activated pixels along each vertical line at different distances by the total number of activated pixels with the given SF. To evaluate the response weights of V4 lateral and medial parts in different SFs (Figure 5B, C) at each single SF condition, we calculated the ratio of the activated pixels in lateral and medial parts to all the activated pixels for a given SF.

After merging all the orientation and color selective pixels at different SF conditions (at least 3 SFs, low: 0.25-0.5 cycles/deg., medium: 1-2 cycles/deg., and high: 3-6 cycles/deg.), we obtained the regions representing nearly the entirety of orientation/color domains (see example in Figure 4F, G) and calculated the coverage ratio of the functional domains within the area (values in Figure 4F, G).”

Lines 168-178：

“We summarized our data and provided the population tendency of how activated functional domains (orientation and color domains) change in fovea vs. parafovea (see Figure 5). For each SF, we divided the number of activated pixels in the lateral or medial parts (see Figure 5—figure supplement 1) by the total number of the activated pixels in this SF (pixels selectively activated in a single SF). We found that the proportions of activated functional domains in the lateral and medial parts of V4 change with SFs in distinct ways. For both orientation and color domains, the proportions increase as SF increases in the medial part and decrease as SF increases in the lateral part. These findings also indicate a difference in SF preference between fovea and parafovea.”

3. The third and most important claim is that spatial frequency and orientation are mapped orthogonally (and recursively) in V2 and V4, as seen in Figure 5 and the Figure 5 supplement. Together these figures present two regions in V4 and two regions in V2. If these are the only analyzed regions, the authors need to specify more clearly how they were selected. Presumably, though, other regions were analyzed, and the authors should present results from all analyzable regions, and use statistical analyses to establish significance.

We thank the reviewers for this suggestion to improve the clarity of selected regions. As reported in previous studies (Tanigawa et al., 2010; Li et al., 2013; Hu et al., 2020) and shown in this paper (Figure 4), not all regions are orientation selective. Here we chose the regions with strong orientation selectivity (see New Figure 6 and Figure 6—figure supplement 2, intersection angles in regions with high orientation selectivity, normalized orientation selectivity>0.5, were used for this analysis). We apologize that we made a mistake in calculating the histogram of the intersection angle. Intersection angles larger than 90 degrees were included in the 80-90 degrees group in the previous analysis, which causes the sharp peak at 80-90 degrees. We have corrected it in the manuscript: for the intersection angles larger than 90 degrees, we did the following transformation, 180-angle, to get the complementary acute angle.

To demonstrate that the intersection angles are more frequently detected at a large angle, we divided the detected intersection angles into three groups (small: 0°-30°, medium: 30°60°, large: 60°-90°) and compared the distribution difference among these groups. The results indicate more (percentage value) 60°-90° intersection angles than other intersection angles. The distribution of the large-angle group is significantly higher than the small and medium groups. In addition, we have compared the distributions between strong orientation selectivity and weak orientation selectivity: the difference is also significant. The statistical information has been added in Figure 6V and the Results section. Lines 234-260:

“Relationship between SF and orientation maps in V4 and V2

Having obtained orientation and SF preference maps from the same cortical region, it becomes possible to analyze the spatial relationships between these maps. We chose V4 regions, which showed strong selective orientation responses (normalized orientation selective values larger than 0.5, see Figure 6B and Figure 6—figure supplement 2B). We determined the iso-orientation and iso-SF contours based on the smoothed orientation angle preference map and SF preference map (see one example in Figure 6—figure supplement 1, 18 iso-orientation gradient contours and 5 iso-SF gradient contours). In V2 and V4, the isoorientation and iso-SF contours predominantly intersect in large angles (see Figure 6G, K, O, and Figure 6—figure supplement 2G, K). To demonstrate that the intersection angles are more frequently detected at a large angle, we divided the detected intersection angles into three groups (small: 0°-30°, medium: 30°-60°, large: 60°-90°) and compared the distribution difference among these groups. The results indicate that there are more (percentage value) 60°-90° intersection angles than other kinds of intersection angles (60°-90° Strong Ori: mean=13.56%, SD=1.87%, N=15; 0°-30° Strong Ori: mean=9.19%, SD=1.63%, N=15; 30°-60° Strong Ori: mean=10.58%, SD=1.45%, N=15; 60°-90° weak Ori: mean=10.29%, SD=1.62%, N=6). The distribution of the large angle group is significantly higher than the small (Wilcoxon range test, p=1.60×10^-5^, n=15 from 5 regions, 2 V2 regions, and 3 V4 regions) and medium groups (Wilcoxon rank sum test, p=1.73×10^-4^). In addition, we compare the distribution between groups with strong orientation selectivity and weak orientation selectivity (see Figure 6P-U). The difference is also significant (large angle group, strong orientation selectivity n=15 from 5 regions, 2 V2 regions, and 3 V4 regions, weak orientation selectivity n=6 from 2 V4 regions; Wilcoxon rank sum test, p=0.0057, see Figure 6V).”

4. The fourth claim is that color-sensitive regions in V4 are more associated with low spatial frequencies. The one significant example (the analysis and statistical tests need to be explained), shown in Figure 6, shows a weak relationship to color for both spatial frequency bands, and the other examples presented in the supplementary are not significant and have even lower absolute relationships. These results, if presented, should be considered inconclusive.

Thanks for raising this comment. We realized that our descriptions needed to be more explicit, we added more descriptions in the corresponding Methods section. Here we adopted the methods reported in Lu’s paper (Lu et al., 2018, Neuron) to classify the high and low SF preference domains. We now describe how we calculated the overlapping between SF and color domains in the method section. Lines 577-580:

“The coverage ratio (e.g., the values at the corners in Figure 4 A, B) was calculated by dividing the number of the selectively activated pixels recorded at each SF by the total number of pixels in the given area.”

Although the proportion of low SF preference in color domains is not very high, this proportion is still higher compared with high SF preference. We agree with the reviewers that the results are not inclusive and have reorganized the corresponding Result section (see Figure 7A-D and Figure 7—figure supplement 1) and revised the Discussion section. Lines 425-428:

“As shown in our results, orientation domains and color domains do not cover the entire site of the V4 area. Further studies are needed to bridge the results between different imaging studies and depict more details of spatial relationships among these different modules.”

5. The fifth claim is for stripe-like periodicity of spatial frequency representation in V2, related to color tuning. This is supported by ostention to binary maps of spatial frequency tuning in Figure 7 and supplement. Establishing this periodicity would require statistical analysis, and in any case, seems impossible since only a sliver of V2 is visible in these brain surface images, so stripes orthogonal to the V1/V2 boundary (i.e. CO stripes) cannot be distinguished from other patterns of spatial frequency tuning. In fact, Figure 5E and S5I do not appear to have iso-frequency contours biased toward that orientation.

Thanks for raising this question. We agree with reviewers that further evidence is needed to support the periodicity of spatial frequency representation in V2. However, according to previous studies (Silverman et al., 1989; Lu et al., 2007) and our results, as shown in new Figure 7 with 2 supplements, color and low SF domains tend to share similar cortical locations in the primate visual system. Due to the sliver of exposed V2, we cannot test whether there is a bias in iso-SF contour orientation (more iso-SF contours lay in the orientation parallel or orthogonal to the V1/V2 border). However, our current V2 data (Figure 7—figure supplement 2: V2 SF preference map obtained for a wide range of higher SFs vs. lower SFs) still support the idea that there is an apparent variation in SF representation in V2 (Figure 7, V2 panels).

As the reviewers suggested, we have now shortened the space and adjusted the diction in the Results section and Discussion section to present the data as an open question requiring further studies and analyses.

Lines 289, 297, 320: delete the word “periodic.”

Lines 323: Revise“Thus, there are SF preference differences in V2 that vary according to a unique periodicity” into “Thus, SF preference differences vary uniquely within V2”.

Lines 383-389:

“SF preference change in V2

We explored whether SF is spatially organized similarly to other attributes in V2. We found that similar to color selective response, SF preference changes within the exposed V2 area, forming different SF preference patches, which supports a general functional layout for SF coding in the visual system (preference for low SF in V1 blob, V2/V4 low SF domains; preference for high SF in V1 interblob, V2/V4 high SF domains).”

Lines 429-432:

“Due to the limit of ISOI, we only yielded a sliver of V2 areas. More evidence and new techniques (e.g., ultra-high field fMRI) are required to test whether the SF preference in V2 changes periodically as other features (e.g., color, orientation, direction, disparity).”

6. I recommend emphasizing the conclusion that spatial frequency and orientation are mapped orthogonally in V2 and V4, and presenting analyses of all analyzable imaged regions, with statistical tests.

We appreciate the opportunity to refine the key findings regarding the orthogonal mapping of SF and orientation in V4. We have presented analyses of all the chosen regions shown in figures (see New Figure 6 and Figure 6—figure supplement 2) with Wilcoxon rank sum test. Also, see the response to Essential Revision point 3.

7. The analysis of spatial frequency selectivity in V1 vs. V2 vs. V4 should include a statistical test. The authors might consider a newly available explanation for lower frequency tuning in V4, that the predominant fraction of V4 neurons is tuned for 3D shape from shading, which by its nature is low in spatial frequency (Srinath et al., Current Biology, 2021). This tuning is mapped into patches that are intercalated with patches tuned for 2D shapes, and the authors could consider how this 2D/3D mapping relates to the mapping they observe for spatial frequency and orientation.

We have realized that more data statistics are needed, and we thank reviewers for their suggestions. We now have provided the statistical test of SF selectivity in V1 vs. V2. vs. V4 (see the response to Essential Revision point 1).

It is an exciting research direction to associate the low SF preference to 2D/3D shapes representation. As the reviewer suggested, we also added a couple of paragraphs in the Discussion section on the possible contribution of low SF preference to 3D shape representation of V4.

Lines 421-428:

“In V4, the predominant fraction of V4 neurons are tuned for 3D shape from shading, which is low in SF (Srinath et al., 2021) and may have a close relationship with our findings that most of V4 prefer low SF. This 3D tuning is mapped into patches intercalated with patches tuned for 2D shapes (including different orientations). As shown in our results, orientation domains and color domains do not cover the entire site of the V4 area. Further studies are needed to bridge the results between different imaging studies and depict more details of spatial relationships among these different modules.”

8. If the authors want to present an analysis of spatial frequency as a function of laterality, a more continuous analysis like marginal response histograms for spatial frequency along the ml axis would be more informative and amenable to statistical analysis.

Thanks for this suggestion. We now have added an analysis of the spatial frequency along the M-L axis. Please also see the response to Essential Revision point 2.

9. I recommend eliminating the analyses of and claims about color vs. spatial frequency and stripe-like periodicity because the data do not seem adequate for testing the hypotheses.

Thanks for this suggestion. Due to the sliver of exposed V2, we agree that our current data are not adequate for claims about color vs. SF and stripe-like periodicity. We now shorten the corresponding result sections and present the data as an open question requiring further studies and analyses. Also, see the response to Essential Revision point 5.

10. The first concert has to do with the use of just two orientation values, 45 and 135 degrees to form a neutral orientation control condition (a cocktail blank), where (I assume because the details were sketchy) responses are added, and then subtracted from the comparison condition with a different color or spatial frequency. Surely, horizontal and vertical orientation regions of the cortex will not be stimulated, or weakly at best, and it will be harder or impossible to determine the preferences of these regions for other parameters, such as SF or color. If the authors are sure this is not a problem I think it needs to be addressed directly, early on in the paper.

We thank reviewers for remaindering us that the robustness of spatial frequency mapping needs to be clarified early in the article, despite limited orientation control. We compared the SF preference results acquired by different orientations (45°+135°, 45°, 135°). We found that different orientations do not cause significant differences in the distribution of high SF preference (SF=4 cycles/deg.) and low SF preference (SF=0.25 cycles/deg.). In contrast, different visual areas do (see New Figure 3—figure supplement 1).

Lines 117-123:

“To confirm that the use of just two orientations, 45 and 135 degrees, can detect a complete picture of SF preference in the imaged area, we compared the SF preference results acquired by different orientations (45°+135°, 45°, 135°) (See Figure 3—figure supplement 1). We found that different orientations do not cause significant differences (two-way ANOVA, p>0.05) in the distribution of high SF preference (SF=4 cycles/deg.) and low SF preference (SF=0.25 cycles/deg.). In contrast, different visual areas do (two-way ANOVA, p<0.05).”

11. The second concern has to do with the distribution of iso-parameter contour intersection angles (e.g. Figures5d and h). The results seem, literally, too good to be true. Previous studies (e.g. Obermeyer and Blasdel, 1993; Hubener et al., 1997) have all shown very much broader distributions. It makes me wonder if the cocktail blank problem above has resulted in some kind of systematic distortion of the SF map, such that the intersection angles are biased (I do not have a more developed perspective on it than that). The second possibility is some kind of numerical error and I would urge the authors to check their code – e.g. try the same calculation with an OR map from one animal superimposed on an SF map from another and see if the expected random distribution is found.

Thanks for this question. We checked our program. We apologize for the mistake in calculating the distribution of the intersection angle. All angles larger than 90° were counted as 90°. Now we have corrected the codes and all relative analyses. Although there is no sharp peak at 80°-90°, the large intersection angles (60-90°) were detected more frequently than other angles (Also see the response to Essential Revision point 3 and New Figure 6 and New Figure 6—figure supplement 2).

12. I will add that the finding of decreasing SF preference with increasing distance from the fovea seems unremarkable and I would suggest de-emphasising it as a main result. It is certainly relevant to interpreting the observations of course because SF can only be interpreted with respect to eccentricity. But it is hardly a new or unexpected finding. Also, the fact that color processing is restricted to low SF values is well established in the color literature – this is probably an unavoidable consequence of the distribution of red and green cones across the retina. Some reference to these long-established results might be made.

Thanks for raising this question. In the new results, we chose two distinct regions, one in the fovea and one far from the fovea (parafovea), for further analysis. These two areas are distinct in their responses to different SFs (see New Figure 5B, C). Instead of showing response amplitude differences among different SFs, our results presented how SF affects the orientation/color selective responses in fovea and parafovea at a functional domain level. The reviewer mentioned that color processing is restricted to low SF values. However, there is a controversy in V2 studies. Some studies showed that there is no clear relationship (e.g., Lu et al., 2018) between SF preference and V2 stripe (most color selective neurons locate in V2 thin stripe), while another study indicates there is a correlation (e.g., Lu et al., 2007). Our results support that there is such a correlation in V2.

13. I found the proposed hypercolumn architecture in Figure 1B very difficult to understand. SFs vary in a continuum, so why are only two levels (low SF and high SF) shown in two different colors? Iso-orientation and Iso-SF lines could have been shown in different colors also (say HSV colors for orientation to show the circular mapping and gray colormap for SF going from low to high). Similar to what has been done for iso-hue and iso-brightness lines in the color region. Perhaps it may be worthwhile to show the same proposed architecture in V1 as well, in which orientation maps form pinwheels and colors are in separate blobs. It was unclear to me how this architecture could have pinwheels/blobs as well.

We have revised the marked colors in this model (see New Figure 1B). This proposed model is based on the results from V2 and V4. The imaged V1 covers an eccentricity of 0-1 degrees. In this imaged region, the preferred SF is within a limited range (3.5-4 cycles/deg., Lu et al., 2018). To map the gradient of SF in this region, more SFs ranging from 3.5 to 4 cycles/deg. should be tested. Based on our current tested SFs, it is hard to describe a detailed gradient change in V1.

Based on our data from V4 and results shown in Nauhaus et al., 2012, some orientation pinwheel centers locate in low SF domains, while others locate in high SF domains. Based on these known results, we proposed in V1 that the architecture may have a layout as shown in Author response image 1. This architecture requires further tests and correction.

**Author response image 1. sa2fig1:** Illustration of the architecture in V1. Some of the pinwheel centers are in low SF, and some are in high SF. The iso-SF contours (achromatic dashed lines) intersect with iso-orientation contours (red lines connecting two neighboring pinwheel centers) mostly at a right angle. Low SF preference regions (circled by light gray dashed lines) are filled with orientation domains that prefer low SF and color blobs (orange patches).

14. It is not clear to me how the details of the functional maps depend on the choice of stimuli. In single unit studies, typically a large number of orientations and SFs are used to independently map the SF and orientation tuning preferences. In contrast, here only 2 orientations are used in one case to map the color space. Even for mapping the orientation space, only 4 orientations are used. For mapping the color space also, only the hues along the red-green axis are varied (L-M pathway). I understand that some of these choices could be due to the recording modality (imaging), but it would be very useful if the authors could discuss how/if these stimulus choices can affect their results. More details of the stimuli, such as the drift rate of the gratings, and the cie (x,y, Y) coordinates of red and green hues would be useful.

Thanks for raising this question. As the reviewer mentioned, more stimulus conditions will provide more detailed SF and orientation response features. However, for intrinsic optical imaging studies, it is usually hard to record that many conditions due to the imaging time for each condition. Although we used a few conditions, we now show evidence that this will not dramatically influence our SF and orientation/color preference results. 1. The SF preference map of V2 and V4 are similar, using different orientations (see Figure 3—figure supplement 1). 2. The color maps obtained using different orientations are similar (Author response image 2).

For overlapping results between different functional domains, the limited number of tested stimuli may cause an underestimate of the coverage of functional domains. However, as we calculate the ratio of overlapping regions to detected functional domains, the sizes of both areas are underestimated, so the results will not be strongly influenced compared to calculating the actual sizes of overlapping regions.

In our experiment, the temporal frequency of the gratings was fixed to 4 Hz, and corresponding drifting speeds of these SF conditions (0.25, 0.5, 1, 1.5, 3, 4 cycle/deg.) are 16, 8, 4, 2.7, 1.3, 1 deg./sec. The CIE (x, y, Y) coordinates of red and green hues are 0.662, 0.328,40 (Red), 0.320,0.613, 40 (Green). The information has been added in the “Visual stimuli for optical imaging” section.

**Author response image 2. sa2fig2:** Color maps obtained by comparing red/green and white/black gratings with different orientations. A. Used orientations: 45° and 135°. B. Used orientations: 45°. C. Used orientations: 135°.

15. Can you show the iso-contour lines for orientation on the orientation maps also as a supplementary figure to see how well the algorithm works? Figure 5A shows iso-orientation lines on the SF map. The iso-SF contours shown in Figure 5B easily correspond to the colors in the SF map shown in 5A, but I had difficulty mapping the orientation. Also, I was wondering whether the way the comparisons are done to get the maps (for example, in Figure 4, the same 4 stimuli are compared in two different ways to get orientation and color maps) can potentially impose some constraints on those maps. I say this because it is striking to me that almost every red and blue line shown in 5C and 5G appears to intersect orthogonally (as also shown in 5D and 5H).

As the reviewer suggested, we now have provided one more figure supplement on an example that shows the iso-orientation gradient lines for the orientation map and iso-SF gradient lines for the SF map (see New Figure 6—figure supplement 1). Lines 239-242:

“We determined the iso-orientation and iso-SF contours based on the smoothed orientation angle preference map and SF preference map (see one example in Figure 6—figure supplement 1, 18 iso-orientation gradient contours and 5 iso-SF gradient contours).”

16. To me the orthogonality of SF and Orientation contours in Figure 5 was the most striking result. Can you show how this analysis looks for V1? The supplementary figure also shows only V2 and V4.

Thanks for raising this question. For V1, its preferred SF is relatively high at the lateral part. In figure 2, most parts of V1 are blue (which means high SF preference).

The imaged region covers an eccentricity of 0-1 degrees. In this imaged region, the SF preference of V1 is within a limited range (~3.5-4 cycles/deg., Lu et al., 2018). In order to map the gradient of SF in this region, more SFs should be tested. Based on our current tested SFs, it is hard to describe a detailed gradient change in V1. However, based on the previous studies (Silverman et al., 1989 and Nauhaus et al., 2012), we proposed that the spatial-frequency organization in V1 is also well organized and follows a similar way as those in V2 and V4 (Also see the response to Essential Revision point 13).

17. The claim about periodicity is not well quantified. If the authors wish to make this claim, they need to show the Fourier transform of the activation pattern as a function of space and show clear peaks in the spectrum. Also, the authors can perhaps clarify what is the spatial resolution of the imaging technique itself.

Thanks for raising this question. Based on our current data, we cannot directly quantify the periodicity of the SF preference in V2 as there are very few repetition cycles detected along the exposed V2 area. However, our results showed a correlation between SF preference and color selectivity in V2. It is well established that in V2, color selectivity varied periodically across different stripes. Thus, our results indicate that SF preferences in V2 may also vary periodically across different stripes. We agree that our current data are not adequate for claims about stripe-like periodicity. We now shorten the corresponding result sections and present the data as an open question requiring further studies and analyses.

Based on the resolution of ISOI, we cannot achieve the resolution at the cellular level. We cannot further analyze regions containing neurons with complex selectivity (e.g., orientation pinwheel). For these tiny delicate structures, methods (e.g., Two-photon imaging) with a high spatial resolution are required. We clarified this in the Discussion section.